# Rab10 regulates the sorting of internalised TrkB for retrograde axonal transport

Oscar Marcelo Lazo[1,2]*, Giampietro Schiavo[1,2]*

[1]Department of Neuromuscular Diseases and UCL Queen Square Motor Neuron Disease Centre, UCL Queen Square Institute of Neurology, University College London, London, United Kingdom; [2]UK Dementia Research Institute at UCL, London, United Kingdom

**Abstract** Neurons process real-time information from axon terminals to coordinate gene expression, growth, and plasticity. Inputs from distal axons are encoded as a stream of endocytic organelles, termed signalling endosomes, targeted to the soma. Formation of these organelles depends on target-derived molecules, such as brain-derived neurotrophic factor (BDNF), which is recognised by TrkB receptors on the plasma membrane, endocytosed, and transported to the cell body along the microtubules network. Notwithstanding its physiological and neuropathological importance, the mechanism controlling the sorting of TrkB to signalling endosomes is currently unknown. In this work, we use primary mouse neurons to uncover the small GTPase Rab10 as critical for TrkB sorting and propagation of BDNF signalling from axon terminals to the soma. Our data demonstrate that Rab10 defines a novel membrane compartment that is rapidly mobilised towards the axon terminal upon BDNF stimulation, enabling the axon to fine-tune retrograde signalling depending on BDNF availability at the synapse. These results help clarifying the neuroprotective phenotype recently associated to Rab10 polymorphisms in Alzheimer's disease and provide a new therapeutic target to halt neurodegeneration.

**\*For correspondence:**
oscar.lazo@ucl.ac.uk (OML);
giampietro.schiavo@ucl.ac.uk (GS)

**Competing interest:** The authors declare that no competing interests exist.

## Editor's evaluation

This important study, of interest to cellular neurobiologists, uses convincing microscopy methods to show that Rab10 GTPase is a new regulator of neurotrophin receptor trafficking and signaling. Defining how neurons respond to spatial extrinsic cues, such as neurotrophins, and relay this information long-distance to influence transcriptional events is an important topic in neurobiology.

## Introduction

Communication between cells depends on their ability to respond as integrated units to spatial and temporal signalling patterns. The complex morphology of neurons provides an unrivalled model to study how sorting and trafficking of signalling complexes coordinate local signalling at axon terminals and the propagation of messages to the cell body. A clear example is provided by neurotrophic factors secreted by target tissues and sensed by axon terminals, where, among other functions, they regulate local cytoskeletal dynamics to promote or cease axon elongation, induce branching, as well as synaptic maturation and plasticity (*Andres-Alonso et al., 2019*; *Szobota et al., 2019*; *Woo et al., 2019*). At the same time, a population of activated receptors are internalised and targeted to the retrograde axonal transport pathway within signalling endosomes (*Barford et al., 2017*; *Villarroel-Campos et al., 2018*), which propagate neurotrophic signalling towards the cell body, regulating

gene expression, dendritic branching, and the balance between survival and apoptosis (*Du and Poo, 2004*; *Pazyra-Murphy et al., 2009*; *Watson et al., 1999*; *Zhou et al., 2012*). How local and central responses are coordinated across the massive distance from axon terminals to the soma is a crucial and still unanswered question for neuronal cell biology.

Brain-derived neurotrophic factor (BDNF), a member of the neurotrophin family, is widely expressed in the central nervous system, together with its receptor tropomyosin-related kinase B (TrkB). By activating diverse signalling cascades, comprising the phosphoinositide 3-kinase (PI3K)-Akt pathway, mitogen-activated protein kinases (MAPKs), and phospholipase C-gamma (PLCγ), BDNF and TrkB play a critical role in the formation, maintenance, and plasticity of neuronal circuits (*Huang and Reichardt, 2003*; *Minichiello, 2009*). Binding of BDNF at synaptic sites leads to endocytosis of TrkB and entry of the activated ligand–receptor complexes in early endosomes (*Deinhardt et al., 2006*). Whilst part of internalised TrkB recycle back to the plasma membrane, a pool of receptors is sorted to signalling endosomes and engage with the cytoplasmic dynein motor complex, which mediates the transport of these organelles to the soma (*Andres-Alonso et al., 2019*; *Ha et al., 2008*; *Zhou et al., 2012*). Independent lines of evidence indicate that this compartment is generated from early endosomes that then mature into a more specialised organelle escaping acidification and lysosomal degradation (*Villarroel-Campos et al., 2018*). Sorting of TrkB receptors from early to signalling endosomes constitutes the critical regulatory node controlling the intensity of the retrogradely propagated signal; however, to date, no clear mechanism controlling this sorting process has been elucidated.

The Rab family of monomeric GTPases plays a central role regulating post-endocytic trafficking of TrkB. Whilst early endosome formation is regulated by Rab5, maturation and processive transport of signalling endosomes in different neuronal models are controlled by Rab7 (*Bucci et al., 2014*; *Burk et al., 2017*; *Deinhardt et al., 2006*; *Kucharava et al., 2020*). However, it is currently unclear whether or not Rab7 is the only member of the Rab family necessary for the latter process. Because axonal retrograde signalling endosomes appear to be a diverse group of organelles (*Villarroel-Campos et al., 2018*), we hypothesised that other members of the Rab family also contribute to the segregation of TrkB and its sorting to retrograde axonal carriers.

In this work, we specifically focused on Rab10 since it has been shown that Rab10-positive organelles are transported both anterogradely and retrogradely along axons in hippocampal neurons (*Deng et al., 2014*). Previous work from our laboratory using an affinity purification approach to isolate neurotrophin signalling endosomes from mouse embryonic stem cell-derived motor neurons found that Rab10 was significantly enriched in this axonal compartment, which is also characterised by the presence of Rab5 and Rab7 (*Debaisieux et al., 2016*). By manipulating Rab10 expression and activity in hippocampal neurons, as well as analysing the axonal dynamics of Rab10 organelles, we have explored its ability to regulate the sorting of TrkB to the retrograde axonal transport pathway and respond to increasing concentrations of BDNF, adjusting retrograde signalling on demand.

## Results

### Decreasing the expression of Rab10 in neurons

To manipulate Rab10 expression levels in hippocampal neurons, we used a lentiviral system encoding a doxycycline-inducible short hairpin RNA (shRNA). We monitored the effects of this virus on cell viability and expression levels of Rab10 and TrkB at different time points after addition of doxycycline. We observed that 48 hr of treatment consistently halved the number of cells per field compared to control group, whereas incubation for 24 hr did not affect neuronal density (*Figure 1a and b*). Rab10 immunoreactivity appeared significantly decreased at both 24 and 48 hr after addition of doxycycline compared to control lentivirus (*Figure 1a and c*). Finally, we monitored expression levels of TrkB receptors, to confirm that this system allows the study of the trafficking of the endogenous receptor upon Rab10 knockdown. We observed comparable levels of TrkB even at 48 hr of treatment with shRNA *Rab10* (*Figure 1d*).

### Rab10 is required for retrograde TrkB trafficking and signalling

To study axonal TrkB dynamics, we cultured hippocampal neurons in microfluidic chambers, which allow the cellular and fluidic compartmentalisation of axon terminals. After 7 days in vitro, neurons plated in one of the compartments (designated as 'somatic') displayed axons reaching the axonal

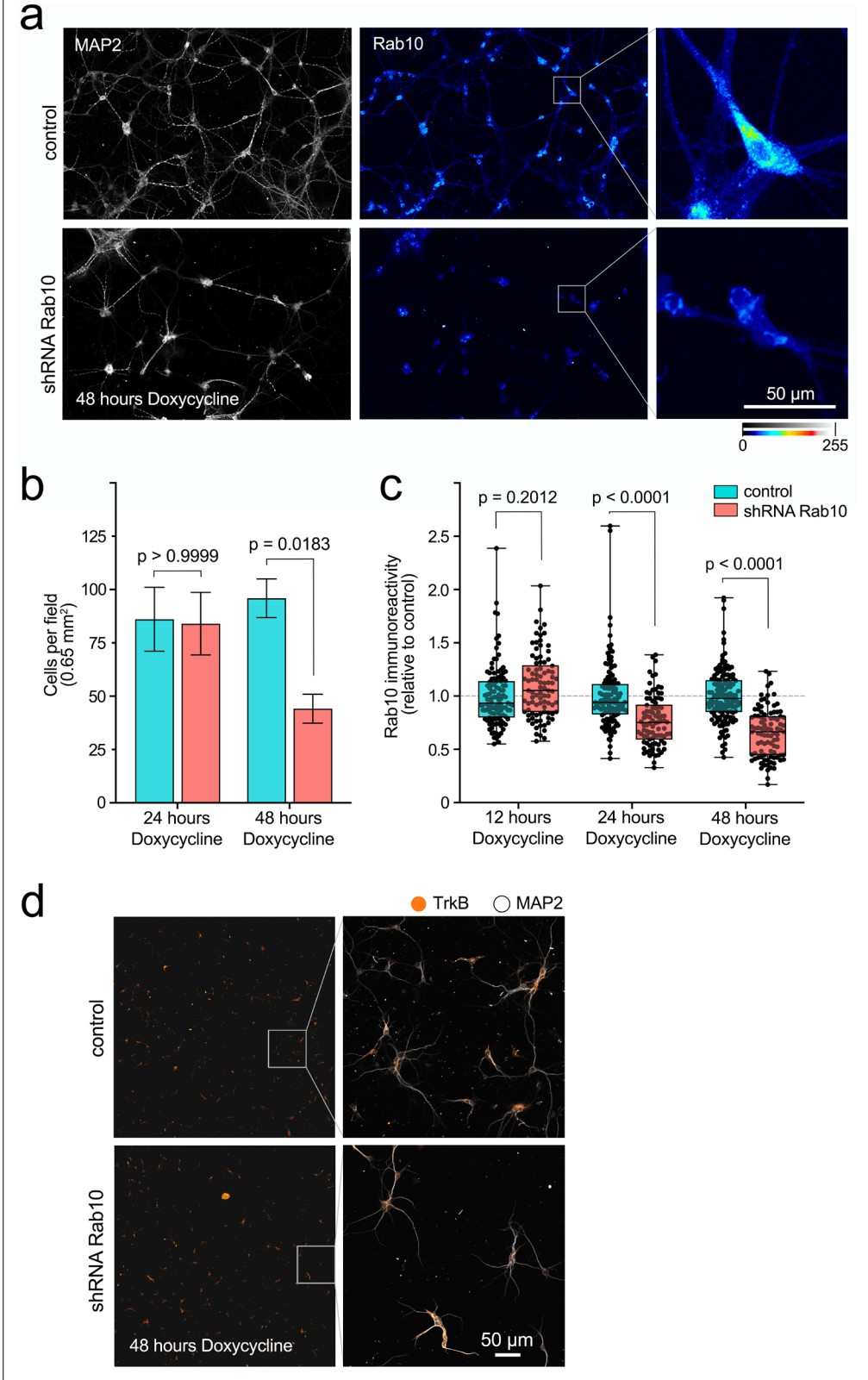

**Figure 1.** Doxycycline-inducible knockdown of Rab10 in hippocampal neurons. (**a**) Representative fields of a primary mass culture of hippocampal neurons transduced with shRNA *Rab10* versus control and treated with doxycycline for 48 hr. Cells have been immunolabelled for MAP2 (grey) and Rab10 (colour scale 0–255). Scale bar = 50 µm. (**b**) Cell density was quantified in 18 fields per treatment across three independent experiments,

*Figure 1 continued on next page*

*Figure 1 continued*

showing a significant decrease after 48 hr. Two-way ANOVA, F(1,68), p value for knockdown = 0.0270, p value for time = 0.0406, p value for interaction = 0.2114 (non-significant). The p values for Bonferroni multiple comparison tests, t(68), are indicated in the plot. (**c**) In the same experiments, immunoreactivity for Rab10 was quantified per cell at 12, 24, and 48 hr with doxycycline, and analysed using two-way ANOVA (p value for knockdown, time, and interaction <0.0001); p values for Bonferroni multiple comparison tests, t(68), are indicated in the plot and show a significant effect of the shRNA at 24 and 48 hr. (**d**) Representative low-magnification fields showing no difference on immunoreactivity for TrkB in hippocampal neurons treated with shRNA *Rab10* versus control after 48 hr with doxycycline. The right panel shows zoomed boxes with TrkB (orange) and MAP2 (grey). Source data of the plots have been included in *Figure 1—source data 1*.

The online version of this article includes the following source data for figure 1:

**Source data 1.** Data tables for each plot presented in *Figure 1* are given as individual CSV files.

compartment (*Figure 2a*). Maintaining a higher volume of media in the somatic compartment allows a micro-flow along the grooves, keeping any substance added to the axonal compartment confined. Using this experimental set-up, we incubated axon terminals with an antibody against the extracellular domain of TrkB and induced its endocytosis by adding 20 ng/mL BDNF. After 2.5 hr, we were able to detect axonal TrkB in the cell body of neurons stimulated with BDNF, but not in neurons depleted of BDNF by addition of an anti-BDNF-blocking antibody (*Figure 2b*), confirming the specificity of the antibody and the overall reliability of our retrograde accumulation assay.

Neurons transduced with the shRNA *Rab10* and treated with doxycycline for 18–22 hr showed lower levels of endogenous Rab10 (*Figure 2c*, grey) and significantly reduced retrograde accumulation of TrkB compared with neurons expressing control lentivirus (*Figure 2c and d*). Since expression of Rab10 is variable among neurons, we tested the correlation between the levels of endogenous Rab10 with the retrograde accumulation of TrkB (*Figure 2e*). We found that, even though there was overlap, control and knocked down neurons clustered as expected. Moreover, when we take both populations together, we found a significant correlation between Rab10 expression and TrkB accumulation, strongly suggesting that Rab10 plays a role in retrograde axonal transport of internalised TrkB (*Figure 2e*).

To confirm the functional consequences of this decrease on the retrograde transport of TrkB, we treated neurons in the axonal compartment with BDNF and analysed the levels of phosphorylated cAMP response element binding protein (pCREB) in the nucleus. pCREB is a well-established proxy for neurotrophic signalling in neurons and has been shown to be critical for global neuronal responses to neurotrophins, such as BDNF-induced dendritic branching (*González-Gutiérrez et al., 2020*). Neurons treated with an shRNA directed against Rab10 showed a significant decrease in nuclear pCREB (*Figure 2f and g*), which was rescued by treatment of Rab10 knockdown cells with a lentivirus encoding shRNA-resistant *Rab10*, further confirming the specificity of this effect.

## Rab10 associates transiently to TrkB-containing retrograde carriers

Given its critical role regulating retrograde accumulation of TrkB, we investigated whether Rab10 was present in retrograde signalling endosomes. As stated before, signalling endosomes are likely to be a heterogeneous pool of functionally related organelles with diverse molecular compositions (*Villarroel-Campos et al., 2018*). Most of these axonal carriers are positive for Rab7 (*Deinhardt et al., 2006*), which enables the recruitment of the dynein motor complex and their processive retrograde transport along microtubules (*Ha et al., 2008*; *Zhou et al., 2012*). Super-resolution radial fluctuations (SRRF) microscopy was used to examine the distribution of endogenous Rab10, Rab7, and Rab5 along the axon of neurons stimulated with 20 ng/mL BDNF for 30 min (*Figure 3a*). We found that Rab10 (orange) and Rab7 (green) consistently showed a very low degree of overlap (see yellow regions in *Figure 3a* and superposition of intensity peaks in *Figure 3b*). On the other hand, the super-resolution imaging revealed a population of organelles where Rab10 (orange) and Rab5 (purple) partially co-localise (see pink regions, *Figure 3a*), suggesting that Rab5-positive retrograde carriers (*Goto-Silva et al., 2019*), or stationary early endosomes, could also contain Rab10. To quantitatively assess whether co-localisation between endogenous Rab10 and these endosomal markers depended on BDNF signalling, we used confocal microscopy to evaluate co-localisation by using Manders index in neurons stimulated with BDNF (20 ng/mL for 30 min), as well as in neurons depleted of BDNF by

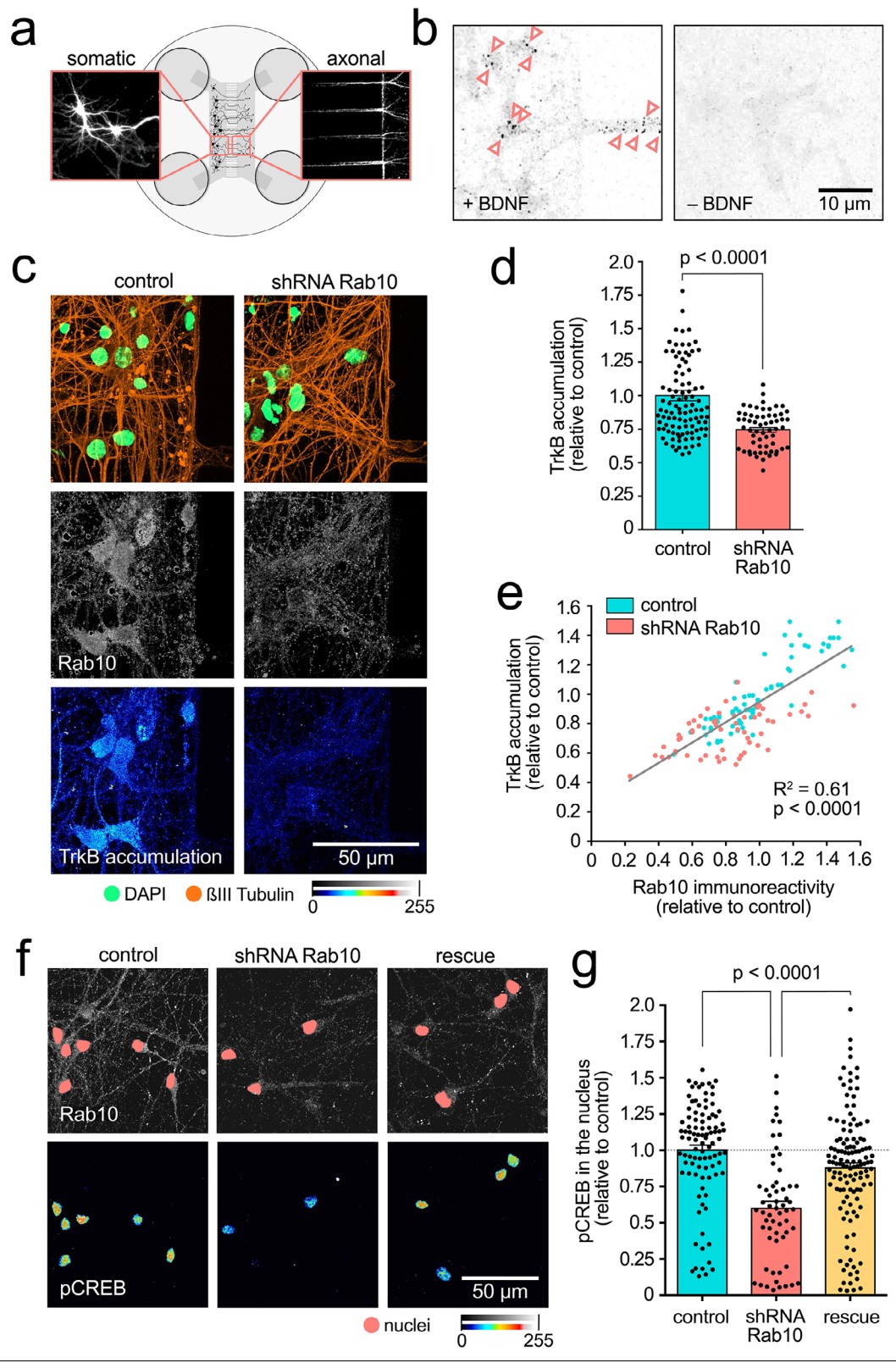

**Figure 2.** Rab10 is required for retrograde TrkB trafficking and signalling. (**a**) Schematic of two-compartment microfluidic chambers highlighting compartmentalisation of somata (left) and axon terminals (right). Micro-flow from somatic to axonal compartments provides fluidic isolation of the somatic compartment to probes added to the other chamber. (**b**) Representative images from the cell bodies of neurons incubated for 2.5 hr with anti-TrkB

*Figure 2 continued on next page*

*Figure 2 continued*

in the axonal compartment, with (+) or without (-) brain-derived neurotrophic factor (BDNF). Pink arrowheads indicate examples of retrogradely transported TrkB-positive organelles. Scale bar: 10 µm. (**c**) Neurons treated with the shRNA targeting *Rab10* were compared to control transduced neurons. Immunofluorescence revealed similar neuronal density (see ßIII-tubulin in orange and nuclear staining in green, top panel), but a decrease in both, expression of Rab10 (grey, middle panel) and retrograde accumulation of TrkB after 2.5 hr (colour intensity scale, bottom panel). Scale bar: 50 µm. (**d**) Quantification of retrograde TrkB accumulation in three independent experiments show statistically significant differences (unpaired Student's *t*-test, t(140), p<0.0001). (**e**) Correlation between expression level of Rab10 and retrograde TrkB accumulation in control and Rab10-knockdown neurons show a significant linear correlation (goodness-of-fit $R^2$=0.61; Pearson r, XY pairs = 131, p<0.0001). (**f**) Axonal stimulation with BDNF for 2.5 hr leads to robust appearance of phosphorylated CREB in the nucleus of control neurons (left panel). This response was impaired in neurons depleted of Rab10 (middle panel), and rescued by the co-expression of a shRNA-resistant mutant Rab10 (right panel). Immunofluorescence for Rab10 is shown in grey, with the nuclei indicated with a pink mask, and nuclear phosphorylated CREB is shown in a colour intensity scale. Scale bar: 50 µm. (**g**) Quantification from three independent experiments showing the statistically significant effect of manipulating Rab10 expression on the levels of phosphorylated CREB in the nucleus (one-way ANOVA, $F_{(2,280)}$, p<0.0001; p values for the Bonferroni multiple comparison tests, t(280), are indicated in the plot). Source data of the plots have been included in *Figure 2—source data 1*.

The online version of this article includes the following source data for figure 2:

**Source data 1.** Data tables for each plot presented in *Figure 2* are given as individual CSV files.

treatment with a blocking antibody. Significance compared to randomised distributions was determined using confined-displacement algorithm (CDA) (*Ramírez et al., 2010*). Rab10 and Rab5 were confirmed to have around 20% co-localisation and a significant increase in the amount of Rab5 in Rab10 domains when stimulated with BDNF (*Figure 3c*). On the other hand, Rab10 and Rab7 were found to exhibit very low co-localisation levels, which was unaffected by BDNF stimulation, suggesting that Rab10 is more likely to be associated with early components of the endosomal system rather than mature and processive signalling endosomes. Similar co-localisation analysis has been done for the recycling endosomes markers Rab4 and Rab11 and included as supplementary material (*Figure 3—figure supplement 1*).

To specifically identify retrograde signalling endosomes containing TrkB, we took advantage of the property of the non-toxic carboxy-terminal domain of the heavy chain of tetanus neurotoxin ($H_CT$) of being transported almost exclusively in retrograde axonal organelles in neurons (*Deinhardt et al., 2006*). We co-internalised $H_CT$-Alexa Fluor647 and mouse anti-Flag antibodies in neurons transfected with a TrkB-Flag construct, and stimulated them with BDNF. Axonal double-positive TrkB/$H_CT$ puncta were considered signalling endosomes moving in the retrograde direction. We analysed the proportion of these organelles co-localising with endogenous Rab10. *Figure 3e* represents the relative area of TrkB/$H_CT$ and the triple-positive area of TrkB/$H_CT$/Rab10 across three different time points (30, 60, and 90 min) post-endocytosis. As expected, the amount of TrkB in retrograde carriers increases with time (cyan). However, the proportion of TrkB present in Rab10 compartments is very low and remains constant across the duration of the experiment (white). These results provide another evidence suggesting that TrkB localises transiently on Rab10-positive membrane compartments *en route* to its delivery to axonal retrograde carriers. This observation is also supported by super-resolution images (*Figure 3f*), showing an increase in TrkB/$H_CT$ double-positive puncta (cyan arrowheads), but very few TrkB/$H_CT$/Rab10 triple-positive puncta (white empty triangles) at 60 and 90 min.

## Overexpressed Rab10 is co-transported with retrograde TrkB

The data presented so far support a model in which Rab10 is critical for retrograde transport of TrkB, but it does not specifically define a stable population of retrograde carriers. Rather, the evidence shown in *Figure 3e and f* suggests that TrkB is transiently associated to Rab10-positive membranes, opening the possibility that Rab10 participates in the sorting of internalised receptors to retrogradely transported signalling endosomes. To directly monitor this process, we overexpressed EGFP-Rab10 in primary hippocampal neurons. Since it is well documented that overexpression of Rab GTPases promotes their activity due to their increased association to membranes (*Zhen and Stenmark, 2015*), we expected that an increase in the abundance and/or activity of Rab10 would stabilise its association to TrkB retrograde carriers, allowing its visualisation by time-lapse microscopy. Neurons were

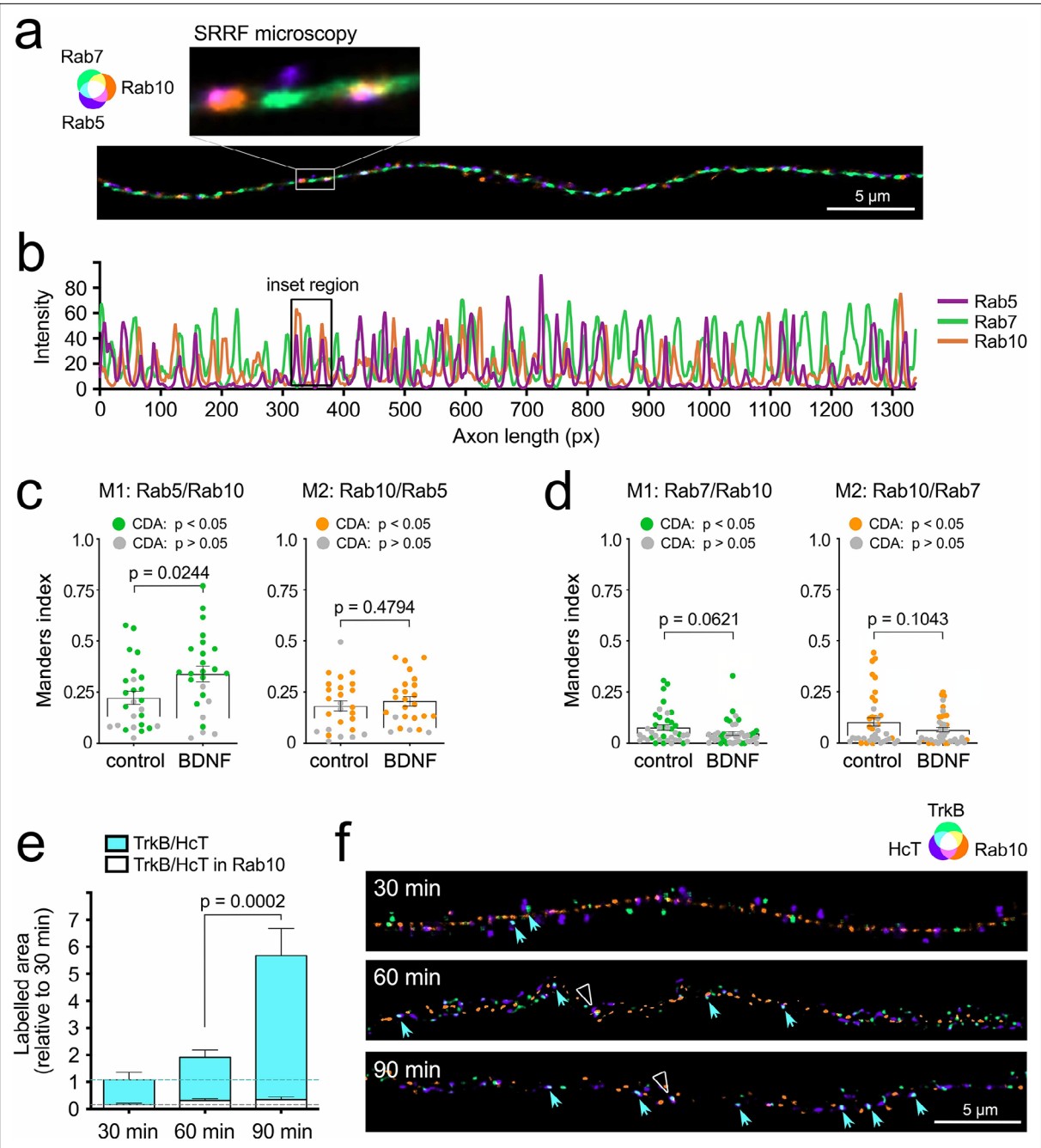

**Figure 3.** Internalised TrkB transiently co-localises with Rab10 in axons. (**a**) Representative super-resolution radial fluctuations (SRRF) microscopy of an axon stained for endogenous Rab5 (purple), Rab7 (green), and Rab10 (orange). The inset at ×6 higher magnification shows examples of partial co-localisation of Rab10 with Rab5 and Rab7. Scale bar = 5 µm. (**b**) Intensity along the same axonal segment shown in (**a**) was plotted to show the correlation between the three markers. (**c**) Co-localisation of endogenous Rab10 and Rab5 was analysed comparing axons from starved neurons versus incubated 30 min with brain-derived neurotrophic factor (BDNF). Confined-displacement algorithm (CDA) has been used to compute Manders coefficients (Welch's corrected unpaired Student's *t*-test; M1: t(46.27), p-value = 0.0244; M2: t(48.94), p-value = 0.4794). Data points showing significant co-localisation compared to random distribution are marked with coloured dots. No significant differences were found between starved and BDNF-stimulated neurons. (**d**) Co-localisation of endogenous Rab10 and Rab7 was analysed in a similar experimental set-up (Welch's corrected unpaired Student's *t*-test; M1: t(72.32), p-value = 0.0621; M2: t(62.33), p-value = 0.1043). (**e**) Labelled H$_C$T and anti-TrkB were co-internalised in the presence of BDNF for 30, 60, and 90 min, and their level of overlap with endogenous Rab10 in axons was evaluated using confocal microscopy. Relative areas positive for H$_C$T and TrkB are shown in cyan (normalised to 30 min) and the fraction of the normalised area that was triple positive for H$_C$T, TrkB and Rab10 is plotted in white. Whereas the double TrkB/H$_C$T-positive area significantly increased by 90 min (one-way ANOVA, F(2,72), p-value <0.0001; Bonferroni multiple comparison test p value is shown in the plot), the triple TrkB/H$_C$T/Rab10 surface remained low and fairly constant at all time points

*Figure 3 continued on next page*

Figure 3 continued

(one-way ANOVA, F(2,72), p-value = 0.2730; Bonferroni multiple comparisons test for 60 vs. 90 min, t(72), p-value >0.9999). (**f**) Representative SRRF microscopy from the same three time points. Double-positive puncta for TrkB (green) and $H_C$T (purple) is indicated with cyan arrowheads. Triple-positive puncta of TrkB/$H_C$T (cyan) and Rab10 (orange) are indicated with white empty triangles. Scale bar = 5 μm. Source data of the plots have been included in *Figure 3—source data 1*.

The online version of this article includes the following source data and figure supplement(s) for figure 3:

**Source data 1.** Data tables for each plot presented in *Figure 3* are given as individual CSV files.

**Figure supplement 1.** Co-localisation of Rab10 with Rab4 and Rab11.

co-transfected with EGFP-Rab10 and TrkB-Flag and, after 1 hr of starvation in non-supplemented Neurobasal medium, we internalised Alexa Fluor647-labelled anti-Flag antibodies in the presence of BDNF for 45 min (*Figure 4a*). Time-lapse confocal imaging of axon segments was performed at least 200 μm from the cell body. Sequential frames of a representative movie are shown in *Figure 4b*, where the retrograde co-transport of TrkB and EGFP-Rab10 is indicated with yellow arrowheads. Kymographs (*Figure 4c*) were generated for EGFP-Rab10 (orange) and TrkB-Flag (green). Tracks are shown in the bottom panel, with examples of co-transport indicated by yellow lines. Quantification of five independent experiments confirmed that approximately 60% of retrograde TrkB carriers are positive for Rab10 under these experimental conditions (*Figure 4d*). Interestingly, no anterograde TrkB/Rab10 double-positive compartments were observed, suggesting that TrkB is present in organelles with a strong retrograde bias.

To extend this analysis to the other physiological BDNF receptors, the internalisation of endogenous p75 neurotrophin receptor (p75$^{NTR}$) was monitored in neurons expressing EGFP-Rab10. After depletion of trophic factors, p75$^{NTR}$ uptake was visualised by incubating neurons for 45 min with an Alexa Fluor647-labelled antibody against the extracellular domain of p75$^{NTR}$, in the presence of BDNF (*Figure 4e*). Live-cell imaging of axon segments was done under conditions identical to those used for TrkB. Representative frames and kymographs of EGFP-Rab10 (orange) and internalised p75$^{NTR}$ (green) are shown in *Figure 4f and g*. Interestingly, p75$^{NTR}$ receptor can be found in both retrograde and anterograde Rab10 carriers. On average, 29.6% of anterograde and 15.5% of retrograde p75$^{NTR}$-containing organelles were found positive for Rab10 across five independent experiments (*Figure 4h*).

## BDNF promotes anterograde trafficking of Rab10-positive compartments

In agreement with the bidirectional transport observed for p75$^{NTR}$ carriers (*Figure 4h*), previous work suggested that Rab10-positive compartments travel along the axon in both directions (*Deng et al., 2014*). In light of these results, we hypothesised that the dynamics of axonal compartments containing Rab10 may be responsive to BDNF signalling to direct the sorting of TrkB to retrograde carriers. We therefore examined the axonal transport of Rab10-positive organelles in hippocampal neurons under two opposite conditions: depletion of BDNF using an anti-BDNF blocking antibody, followed by stimulation with 50 ng/mL BDNF (*Figure 5—figure supplement 1*).

*Figure 5a* shows representative frames of an EGFP-Rab10 organelle moving in the retrograde direction along the axon (white arrows) in the absence of BDNF. Five-minute segments of time-lapse microscopy have been used to generate kymographs (*Figure 5b*) at different time points of BDNF stimulation: before BDNF (top), immediately after BDNF addition (centre), and after 10 min of BDNF incubation (bottom). Traces have been colour-coded as retrograde (cyan), anterograde (pink), or stationary/bidirectional (yellow) to reveal changes in the direction bias of Rab10 organelles in the same axon, before, and after the addition of BDNF. Quantification of five independent experiments shows that BDNF-depleted axons exhibit a bias towards retrograde Rab10 transport, which significantly switches to anterograde after 10 min of stimulation with BDNF (*Figure 5c*).

These surprising results reveal a novel mechanism ensuring a tight balance between retrograde and anterograde transport of Rab10 organelles, which is fine-tuned by BDNF. This result predicts that any local increase in BDNF release from post-synaptic compartments will increase the abundance of Rab10 organelles in the immediate vicinity of the source of BDNF.

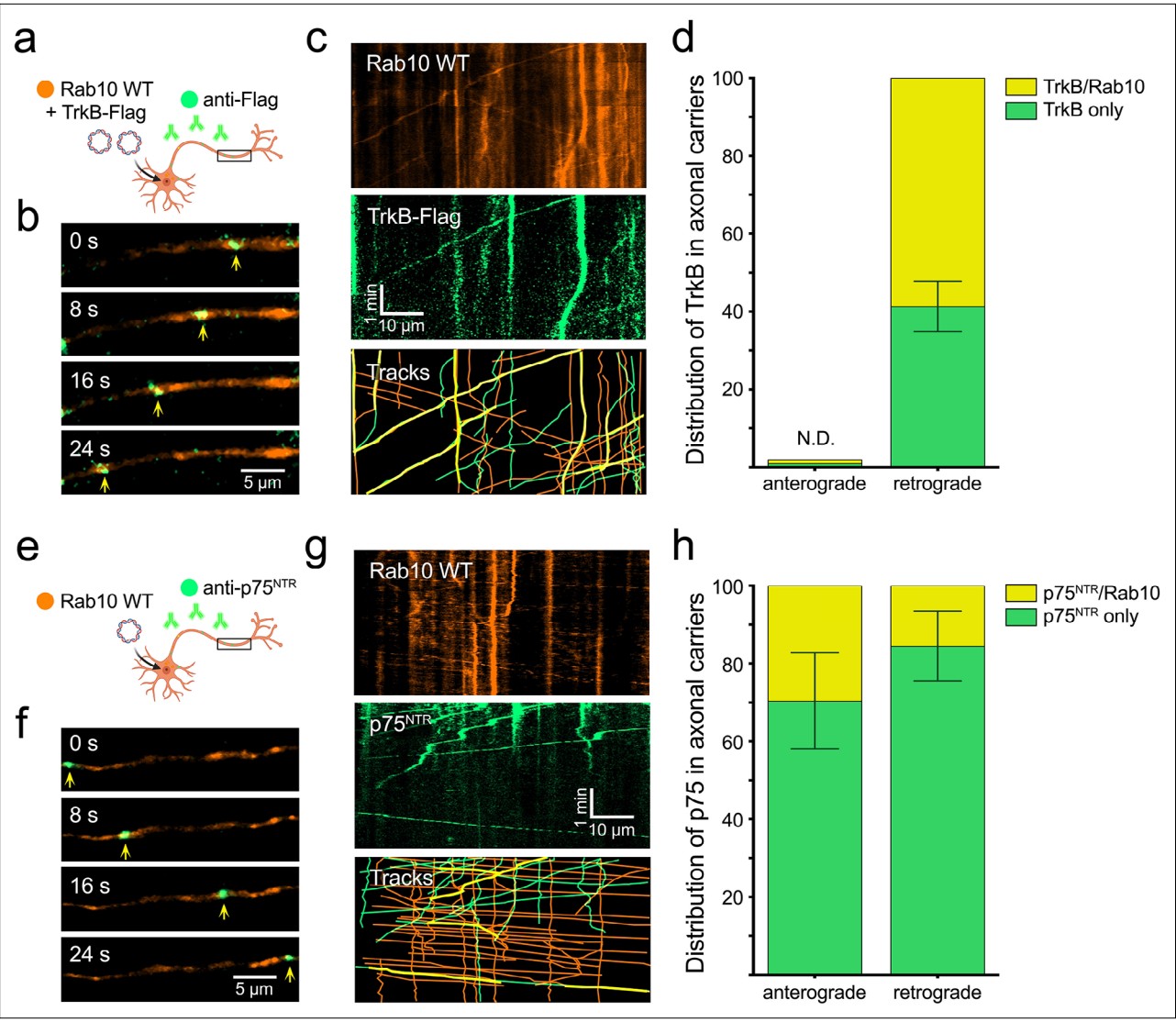

**Figure 4.** Over-expressed Rab10 is co-transported with TrkB and p75 receptors in the axon. (**a**) Hippocampal neurons in mass culture were co-transfected with EGFP-Rab10 and TrkB-Flag plasmids. Fluorescently labelled anti-Flag antibodies were internalised in the presence of brain-derived neurotrophic factor (BDNF) and their axonal dynamics monitored by time-lapse microscopy. (**b**) Representative images of the axon of a double transfected neuron, where a double-positive organelle for EGFP-Rab10 (orange) and anti-Flag (green) is indicated with yellow arrowheads. Scale bar = 5 µm. (**c**) Representative kymograph of the axon of a double transfected neuron during 5 min of imaging showing Rab10 and TrkB-Flag channels. Double-positive tracks (in yellow) show transport predominantly in the retrograde direction (right to left). Scale bar = 10 µm. (**d**) Quantification of five experiments showing the proportion of TrkB-containing mobile organelles that were positive for Rab10. No anterograde TrkB organelles were found; therefore, presence of Rab10 could not be determined (N.D.). (**e**) An equivalent experiment was performed by transfecting EGFP-Rab10 and visualising it together with endocytosed fluorescently-labelled anti-p75$^{NTR}$ antibodies. (**f**) Representative frames from a time-lapse movie displaying a double-positive organelle for Rab10 (orange) and p75$^{NTR}$ (green) moving anterogradely. Scale bar = 5 µm. (**g**) Representative kymograph showing Rab10 and p75$^{NTR}$ channels. Double-positive tracks (in yellow) show transport in both anterograde (left to right) and retrograde directions. Scale bar = 10 µm. (**h**) Quantification of five experiments displaying the proportion of p75$^{NTR}$ mobile organelles positive for Rab10 moving in the anterograde and retrograde direction. Source data of the plots have been included in *Figure 4—source data 1*.

The online version of this article includes the following source data for figure 4:

**Source data 1.** Data tables for each plot presented in *Figure 4* are given as individual CSV files.

## BDNF increases the recruitment of KIF13B to Rab10 organelles in the axon

The uniform polarity of the microtubules in the axon of mature neurons implies that the directionality of axonal carriers largely depends on their association to anterograde or retrograde motor proteins

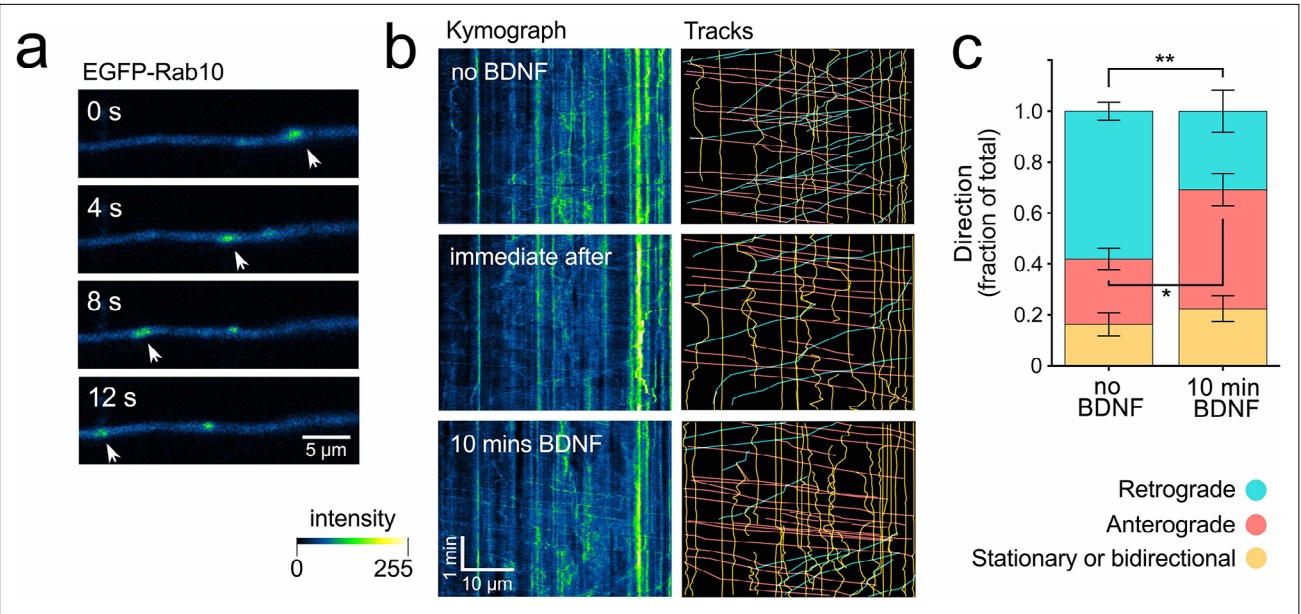

**Figure 5.** Brain-derived neurotrophic factor (BDNF) regulates the directionality of Rab10 organelles. Hippocampal neurons in mass culture were transfected with EGFP-Rab10 and depleted of BDNF for 60 min. (**a**) Representative axon of a live neuron showing retrograde (right to left) transport of a Rab10-positive organelle (white arrowheads) in the absence of BDNF. Scale bar = 5 µm. (**b**) In the panels on the left, representative kymographs (colour-coded as in **a**) are presented from the same axon upon BDNF depletion (top), immediately after the addition of 50 ng/mL of BDNF (middle), and 10 min thereafter (bottom). In the panels on the right, tracks have been traced and categorised as retrograde (cyan), anterograde (pink), or stationary/bidirectional (yellow). Scale bar = 10 µm. (**c**) The frequencies of tracks from each of the three categories have been quantified and plotted, comparing no BDNF and 10 min post-addition of 50 ng/mL BDNF. N = 14 axonal segments from 10 independent experiments. Unpaired Student's *t*-test, t(14), showed a significant increase in anterograde carriers (p-value = 0.0150, *) at the expense of retrograde carriers (p-value = 0.003, **). Stationary and bidirectional carriers did not show any significant change (p-value = 0.4278). See *Figure 5—video 1* for a video. Source data of the plots have been included in *Figure 5—source data 1*.

The online version of this article includes the following video, source data, and figure supplement(s) for figure 5:

**Source data 1.** Data tables for each plot presented in *Figure 5* are given as individual CSV files.

**Figure supplement 1.** Example of a kymograph showing the dynamics of the constitutively-active EGFP-Rab10 Q68L mutant in a representative axon depleted of brain-derived neurotrophic factor (BDNF).

**Figure 5—video 1.** Representative axon in mass culture expressing EGFP-Rab10 (with intensity colour-coded as in *Figure 5a*) before and after being stimulated with 50 ng/mL brain-derived neurotrophic factor (BDNF).
https://elifesciences.org/articles/81532/figures#fig5video1

**Figure 5—video 2.** Representative time-lapse microscopy video of EGFP-Rab10 (with intensity colour-coded as in *Figure 5*) after 10 min upon stimulation with 50 ng/mL brain-derived neurotrophic factor (BDNF).
https://elifesciences.org/articles/81532/figures#fig5video2

**Figure 5—video 3.** Neuron in mass culture expressing EGFP-Rab10 (with intensity colour-coded as in *Figure 5a*) at steady state.
https://elifesciences.org/articles/81532/figures#fig5video3

(*Guillaud et al., 2020*; *Kwan et al., 2008*). To provide further mechanistic insights into the directional switch of Rab10 axonal carriers upon BDNF stimulation, we analysed the distribution of two antero-grade microtubule-associated motor proteins predominantly found in central neuron axons, KIF5B and KIF13B (*Yang et al., 2019*). *Figure 6a and d* show representative high-resolution confocal images of axons where endogenous Rab10 (green) and either KIF13B or KIF5B (orange) have been detected. Neurons were either starved in the presence of a blocking anti-BDNF antibody (control) or treated for 30 min with 50 ng/mL BDNF (BDNF). Insets displaying two different views of three-dimensional reconstructions of axonal segments show Rab10 membranes positive for KIF13B (*Figure 6a*) or KIF5B (*Figure 6d*) in the absence and presence of BDNF. Interestingly, the axonal pool of KIF13B appears to decrease after stimulation with BDNF, whereas that of KIF5B increases under the same experimental conditions, suggesting that the total axonal content of these kinesins may be differentially regulated by BDNF signalling.

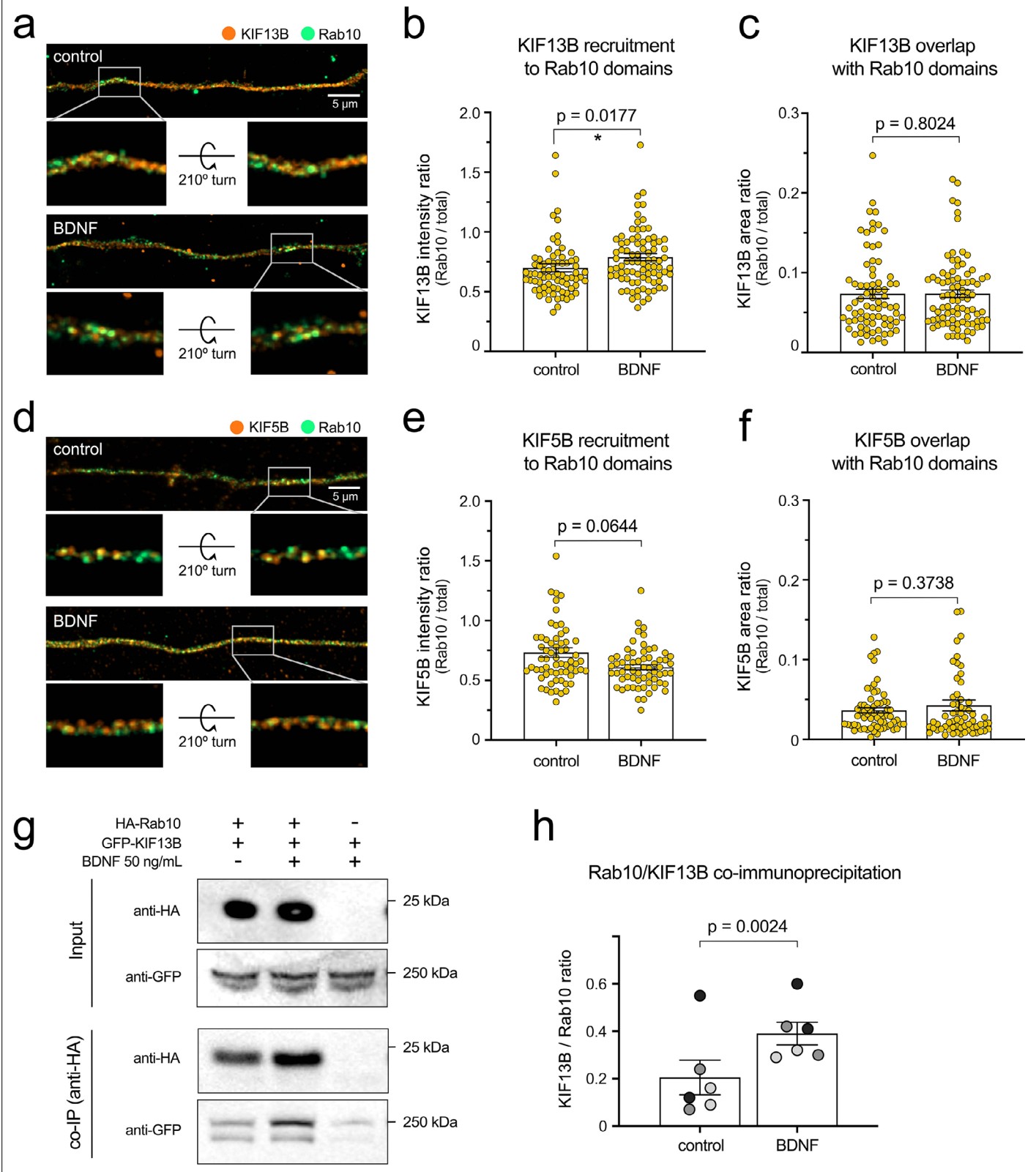

**Figure 6.** Brain-derived neurotrophic factor (BDNF) increases recruitment of KIF13B to Rab10 domains. (**a**) The co-distribution of endogenous KIF13B (orange) and Rab10 (green) was monitored using high-resolution Airyscan confocal microscopy in axons of neurons with or without BDNF for 30 min. Top images correspond to maximum projection of z-stacks, scale bar = 5 μm. A 3D reconstruction of the inset area (grey frame) is shown for each top image on its original position and after turning the image 210° around the x axis. (**b**) KIF13B intensity was measured in the entire axon segment and in

*Figure 6 continued on next page*

*Figure 6 continued*

Rab10-positive areas, and the intensity ratio was plotted and analysed showing a significant enrichment of KIF13B in Rab10 areas upon 30 min of BDNF stimulation (Kolmogorov–Smirnoff nonparametric *t* test, t(79.5), p-value = 0.0177). (**c**) Ratio between KIF13B-positive area that overlaps with Rab10 from total KIF13B area was plotted, finding no difference between neurons starved or incubated with BDNF 30 min (Kolmogorov–Smirnoff nonparametric *t* test, t(79.5), p-value = 0.8024). (**d**) Co-distribution of KIF5B (orange) and Rab10 (green) was also analysed and displayed as in (**a**). Scale bar = 5 µm. Insets show 3D reconstructions on their original and rotated position. (**e**) Quantification of intensity ratio of KIF5B in Rab10 domains versus total KIF5B in the axon shows no significant difference between starved and BDNF-treated neurons (Kolmogorov–Smirnoff nonparametric *t* test, t(62.5), p-value = 0.0644). (**f**) Proportion of the KIF5B-positive area that overlaps with Rab10 is lower than KIF13B and is not altered by BDNF stimulation (Kolmogorov–Smirnoff nonparametric *t* test, t(62.5), p-value = 0.3738). (**g**) The interaction of HA-Rab10 and GFP-KIF13B from N2A cells is modulated by BDNF. Top panel: representative western blot showing the presence of both proteins in the lysate (input). Bottom panel: Western blot of co-immunoprecipitated samples from the same experiment using an antibody against the HA tag. (**h**) Quantification of the ratio between normalised KIF13B and Rab10 in three independent experiments. Western blots have been done in duplicate, and the corresponding paired experiments are indicated by data points of the same shade of grey. Groups were compared using paired Student's *t* test, t(6), p-value = 0.0024. Source data of the plots have been included in *Figure 6—source data 1*.

The online version of this article includes the following source data and figure supplement(s) for figure 6:

**Source data 1.** Data tables for each plot presented in *Figure 6* are given as individual CSV files, as well as unedited representative blots used in *Figure 6g*.

**Figure supplement 1.** Co-immunoprecipitation of HA-Rab10 and GFP-KIF13B.

**Figure supplement 2.** Co-localisation of Rab10 and JIP3/JIP4.

Quantification of the intensity of KIF13B in Rab10-positive areas (*Figure 6b*) reveals a statistically significant 12.7% increase in the amount of the motor recruited to Rab10-positive organelles. On the other hand, *Figure 6c* shows that the ratio between the Rab10-positive and total KIF13B-positive area in the axon remained unchanged (7.3% of the area), suggesting that the proportion of double KIF13B/Rab10 compartments remains constant. In contrast, when we performed the same analysis for KIF5B, we found that, despite its overall increase in the axon upon BDNF stimulation, there is no significant change in the intensity ratio in Rab10-positive domains (*Figure 6e*). Consistently, *Figure 6f* shows that the proportion of KIF5B axonal organelles that overlap with Rab10 is lower than KIF13B (around 4% of the area) and remained unchanged after BDNF stimulation.

Prompted by the results shown in *Figure 6b*, we investigated whether KIF13B and Rab10 interact in a BDNF-dependent manner by expressing GFP-KIF13B and HA-Rab10 in Neuro-2A (N2A) cells. This mouse neuroblastoma cell line expresses TrkB and has been previously used in our laboratory to study trafficking and signalling of neurotrophic receptors (*Terenzio et al., 2014*). After 1 hr of starvation, transfected cells were treated or not with BDNF 50 ng/mL and then we used anti-HA-conjugated magnetic beads to pull down HA-Rab10. Lysates, as well as the co-immunoprecipitated samples eluted from the beads, were analysed by Western blot. A representative example is shown in *Figure 6g*, with the top panels showing the total amount of HA-Rab10 and GFP-KIF13B in the lysates (input). Co-immunoprecipitated HA-Rab10 and GFP-KIF13B eluted from the beads are displayed in the bottom panels (co-IP). Lysates from N2A cells only expressing GFP-KIF13B were used as controls. Quantification of three independent experiments (*Figure 6h*) show an average twofold increase in the KIF13B/Rab10 ratio when cells were stimulated with BDNF. A consistent change in pulled down KIF13B with no change of Rab10 is shown as separated plots in *Figure 6—figure supplement 1*.

## Rab10 regulates the sorting of TrkB in early endosomes, with no effect on recycling

After endocytosis, TrkB accumulates in early endosomes, from which it is sorted either to the recycling route or to endosomal organelles with signalling capabilities (*Deinhardt et al., 2006*; *Zhou et al., 2012*). Rab10, on the other hand, has been shown to regulate multiple processes, including trafficking across early endosomes, the formation of specialised tubular endosomes, the recycling of cargoes back to the plasma membrane, as well as the targeting of plasmalemmal precursor vesicles (PPVs), among other functions (*Babbey et al., 2006*; *Brewer et al., 2016*; *Deng et al., 2014*; *Etoh and Fukuda, 2019*; *Xu et al., 2014*). To understand the mechanism linking Rab10-positive organelles with the retrograde axonal transport of TrkB in signalling endosomes, we designed experiments to discern between a potential role of Rab10 on recycling of internalised TrkB back to the plasma membrane, and sorting of TrkB out of early endosomes into retrograde transport carriers (*Figure 7a*).

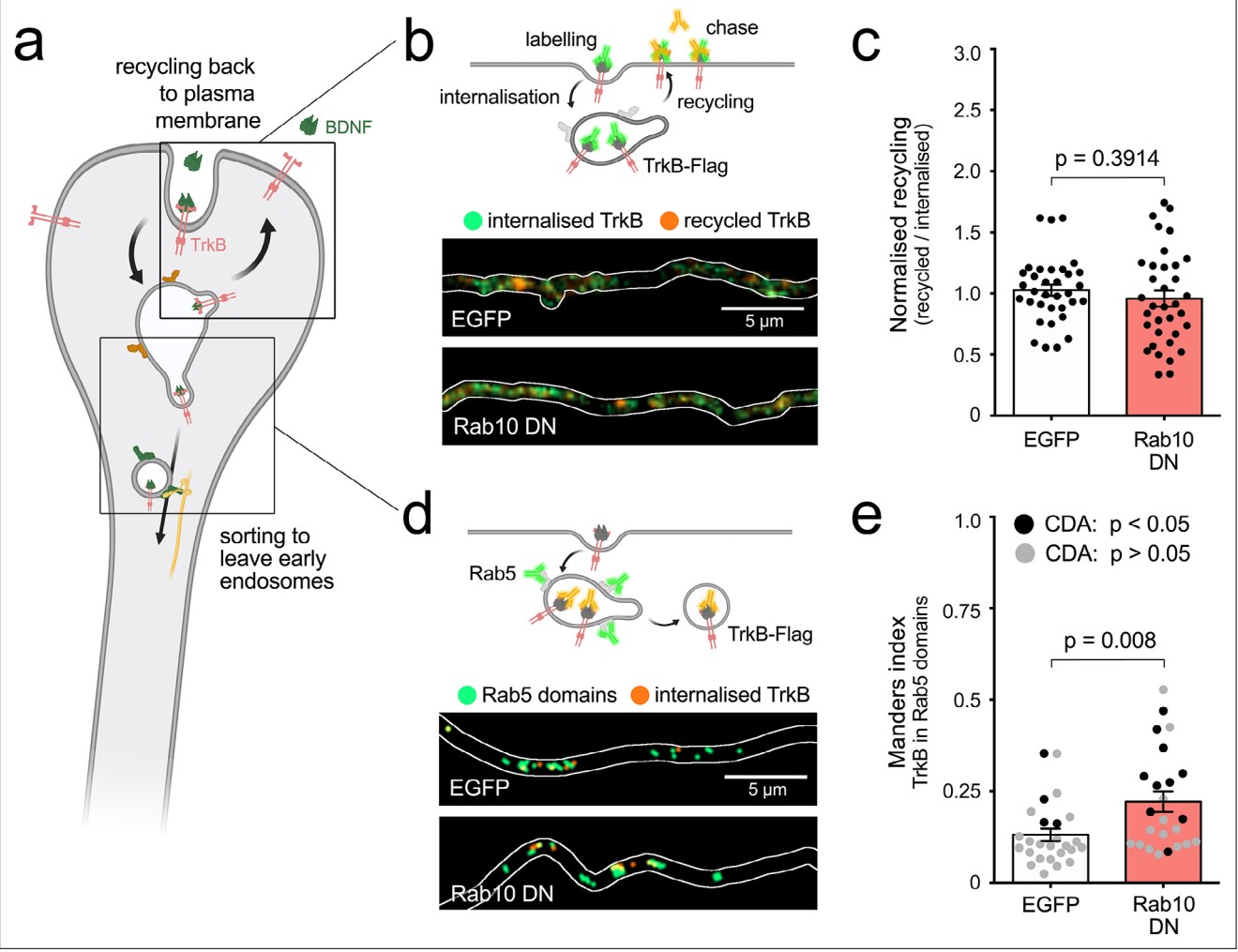

**Figure 7.** Rab10 regulates sorting of TrkB out of early endosomes. (**a**) Main hypotheses about the role of Rab10 regulating the sorting of TrkB to signalling endosomes include recycling back to the plasma membrane or sorting of TrkB receptors out of early endosomes to retrograde carriers. (**b**) Top: diagram of the experiment showing TrkB-Flag receptors on the axonal surface bound to anti-Flag antibodies (green). After internalisation, the remaining anti-Flag is removed from the surface and the labelled receptor that recycled to the plasma membrane is chased with a secondary antibody (orange). Bottom: representative examples of internalised TrkB (green) and recycled TrkB (orange) in axons from neurons transfected with EGFP or a Rab10 DN mutant. (**c**) Normalised recycling shows no difference between EGFP and Rab10 DN transfected neurons. Unpaired Student's *t*-test, t(60.40), p-value = 0.3914. (**d**) Top: diagram of the internalisation of TrkB-flag labelled with anti-Flag antibodies (orange) to Rab5-positive early endosomes (green). Bottom: representative thresholded microscopy images from the axon of neurons transfected with EGFP or Rab10DN mutant. While the amount of orange puncta is similar in both conditions, yellow areas showing co-localisation of internalised TrkB and Rab5 are increased upon Rab10 DN expression. (**e**) Quantification of co-localisation between internalised TrkB-Flag and endogenous Rab5 is significantly higher in neurons expressing Rab10DN compared to EGFP. Unpaired Student's *t*-test, t(38.22), p-value = 0.008. Significant co-localisation according to confined-displacement algorithm (CDA) (p-value < 0.05 compared to randomised signal) is shown with black circles, while inconclusive co-localisation (p-value > 0.05) is shown in grey. Scale bars = 5 µm. Source data of the plots have been included in *Figure 7—source data 1*.

The online version of this article includes the following source data for figure 7:

**Source data 1.** Data tables for each plot presented in *Figure 7* are given as individual CSV files.

To assess the contribution of Rab10 to the recycling of TrkB in the axon, hippocampal neurons were transfected with EGFP or a dominant negative mutant of Rab10 (*Rab10^T23N*; referred to as Rab10DN) and TrkB-Flag. As illustrated in *Figure 7b*, endocytosis of anti-Flag M1 antibodies was allowed for 30 min in the presence of BDNF, and then the residual antibody still bound to the neuronal surface was removed using EDTA, which dissociates this antibody from the Flag peptide (*Chen et al., 2005*). Recycling receptors were then chased using Alexa Fluor647-conjugated secondary antibodies. After fixation, internalised TrkB-Flag was labelled with Alexa Fluor555-conjugated secondary antibodies.

Comparison between recycling ratio (recycled/internalised) of EGFP- and Rab10DN-expressing neurons shows no significant differences (*Figure 7c*).

In contrast, if the sorting of TrkB to retrograde signalling carriers was regulated by Rab10, expression of Rab10DN would result in retention of a population of TrkB in early endosomes, driving an increase of co-localisation between internalised TrkB and Rab5. Therefore, we incubated neurons transfected with TrkB-Flag and either EGFP or Rab10DN, with anti-Flag antibodies for 30 min in the presence of BDNF, and then analysed TrkB/Rab5 co-localisation using Manders index and CDA (*Figure 7d and e*). Rab10DN caused a statistically significant increase of internalised TrkB in Rab5-positive domains from 13.2% ± 1.7 to 22.2% ± 2.8, and an increase in the reliability of the co-localisation measurements (CDA p<0.05) (*Figure 7e*). These results indicate that the probability of finding axonal TrkB in Rab5-positive early endosomes increases in neurons expressing Rab10DN, thus indicating that Rab10 activity modulates the sorting of activated TrkB receptors from axonal early endosomes to retrograde transport organelles.

## Discussion

Our results unravel a novel role for Rab10 in regulating the sorting of internalised TrkB receptors to the retrograde axonal transport pathway. This function appears to be necessary not only for efficient trafficking of TrkB from axons to soma, but also for the propagation of neurotrophic signalling from distal sites to the nucleus, as shown by the decrease of BDNF-induced CREB activation upon Rab10 knockdown (*Figure 2*). CREB is a transcription factor supporting neuronal survival and differentiation, which is activated by phosphorylation downstream Akt and MAPK signalling (*Wang et al., 2018*). Some of the better characterised early response genes to neurotrophic factors (e.g., Egr1, Egr2, Arc, and cFos) are transcriptional targets of CREB, which is also required for BDNF-induced dendritic branching (*Esvald et al., 2020*; *González-Gutiérrez et al., 2020*; *Kwon et al., 2011*). Therefore, CREB phosphorylation is one of the best proxies for global responses to neurotrophic signalling. Long-distance activation of CREB has been reported from distal axons and dendrites, and the endosomal trafficking of phosphorylated TrkB and signalling interactors has been shown to play a crucial role in propagating neurotrophin signalling from the periphery to the nuclear pCREB (*Cohen et al., 2011*; *Riccio et al., 1997*; *Watson et al., 1999*). In our study, we found that Rab10 expression was critical for maintenance and survival of differentiated neurons since its downregulation for 48 hr led to a significant decrease in neuronal density (*Figure 1*). Similar effects have been reported upon overexpression of dominant-negative mutants of Rab5, the GTPase mediating the formation of early endosomes, but not for Rab11, which controls the recycling of receptors to the plasma membrane (*Lazo et al., 2013*; *Moya-Alvarado et al., 2018*), suggesting that these distinct arms of the endosomal network differentially impact on neuronal homeostasis.

Acting as a highly specialised network, Rab GTPases are master regulators of specific membrane trafficking events in eukaryotic cells (*Stenmark, 2009*). Rab10 is one of the few exceptions to this rule, and during the last 30 years, it has been associated to multiple trafficking pathways, including polarised exocytosis from early endosomes, exocytosis in adipocytes and neurons, endoplasmic reticulum dynamics, and the formation of tubular endosomes, to cite but a few (*Chua and Tang, 2018*). This multiplicity of functions is reflected by the diversity of its interactors and effectors, emphasising the importance of the specific context in which Rab10 operates. Here we have confirmed previous findings showing that Rab10-positives organelles are present in axons of hippocampal neurons and undergo bidirectional transport (*Deng et al., 2014*). Importantly, in this work we have demonstrated that the balance between anterograde and retrograde transport of Rab10-positive organelles is regulated by BDNF, indicating the ability of Rab10 to specifically respond to extracellular cues in a signalling context. A similar function of Rab10 balancing anterograde and retrograde transport has been shown to be required for correct dendritic patterning in *Drosophila* (*Taylor et al., 2015*).

We hypothesised that the differential recruitment of members of the kinesin superfamily of anterograde motors could explain a rapid change in the directionality of Rab10 organelles. Given that many kinesins distribute preferentially to distinct neuronal compartments (*Hirokawa and Tanaka, 2015*), we focused on two kinesins that have been found to be predominant in axons, KIF5B and KIF13B (*Yang et al., 2019*). We found that KIF13B, but not KIF5B, increased its relative abundance in axonal Rab10-positive domains (*Figure 6a–f*), which is in line with the recent evidence indicating a direct interaction between Rab10 and KIF13A and B, via a Rab-binding domain (RBD)-homology domain

(*Etoh and Fukuda, 2019*). Moreover, when we used immunoprecipitation to investigate the interaction of KIF13B and Rab10, we found that their binding significantly increases in the presence of BDNF (*Figure 6g and h*). Interestingly, KIF13B and other members of the kinesin-3 family have been shown to yield super processive motion when acting as dimers (*Soppina et al., 2014*), helping to explain how even a relatively small increase in the recruitment of KIF13B onto axonal Rab10-positive organelle can trigger a rapid change in their transport directionality. Together with an increase of the recruitment of KIF13B and other kinesins, Rab10 effectors can also regulate the motor activity of cytoplasmic dynein (*Taylor et al., 2015*) or mediate the interaction with axonal myosins (*Liu et al., 2013*; *Welz and Kerkhoff, 2019*), where specific contribution to directionality and processivity of these organelles will require further exploration.

Although the main focus of this work has been demonstrating a novel role of Rab10 in the retrograde propagation of neurotrophic signalling, the recruitment of KIF13B onto Rab10-positive organelles is an important finding to start disentangling the mechanisms that control the BDNF-dependent switch between retrograde and anterograde transport. For example, the activity of Rab10 is known to be controlled by PI3K-Akt, a canonical BDNF/TrkB signalling pathway. In adipocytes and muscle cells, Rab10 is known to regulate the plasma membrane delivery of the glucose transporter GLUT4 in response to insulin, where activation of Akt leads to phosphorylation of the Rab GAP Akt substrate of 160 kDa (AS160). Rab10 associated to GLUT4-containing endosomes is kept inactive by AS160 until Akt signalling releases the brake and promotes fusion with the plasma membrane (*Sano et al., 2007*).

However, two key observations suggest that directionality of Rab10 organelles in the axon does not depend uniquely on Rab10 activity. First, both anterograde and retrograde organelles have membrane-bound Rab10 (*Figure 5*), which is generally accepted to be its GTP-bound active form. Second, we observed that the constitutively active Rab10$^{Q68L}$ mutant is transported predominantly in the retrograde direction in neurons depleted of BDNF (see *Figure 5—figure supplement 1*). Altogether, these findings indicate that whilst the GTP-bound conformation of Rab10 is necessary to drive its binding to the membrane of axonal organelles, this alone is not sufficient to determine the direction of transport. Signalling molecules associated to the membrane of the moving endosome are likely to be key defining its direction and processivity, as exemplified in *Figure 5—video 2*, which shows an anterograde Rab10-positive carrier running next to a stationary organelle, partially merging and continue moving together.

The LRRK2-dependent phosphorylation of Rab10 in the highly conserved switch II region has been proposed to affect its GTP/GDP cycle as well as its ability to bind effectors (*Pfeffer et al., 1995*; *Xu et al., 2021*). Moreover, this phosphorylation site has been shown to regulate the interaction of Rab10 with JIP3 and JIP4 (*Waschbüsch et al., 2020*), which are adaptors for the plus-end-directed microtubule-dependent motor kinesin-1 (*Cockburn et al., 2018*; *Isabet et al., 2009*). Interestingly, JIP3 has been shown to mediate the anterograde transport of TrkB in neurons and, by that mechanism, to enhance BDNF signalling (*Huang et al., 2011*; *Sun et al., 2017*). However, preliminary experiments showed that JIP3 and JIP4 association to Rab10 organelles is extremely low in hippocampal neurons and does not respond to BDNF stimulation (*Figure 6—figure supplement 2*). LRRK2-phosphorylated Rab10 has been also found to recruit myosin Va (*Dhekne et al., 2021*), providing other candidates for regulation of axonal trafficking. On the other hand, positioning of Rab10-positive membranes has been recently shown to determine their phosphorylation by LRRK2, suggesting that trafficking and composition of these organelles can be mutually regulated (*Kluss et al., 2022*). Whether LRRK2-mediated phosphorylation of Rab10 modulates the interaction with KIF13B is part of our ongoing research efforts and will not be discussed here in detail; however, it is worth mentioning that all three Rab10, KIF13B, and LRRK2 have been implicated in the formation of tubular endosomes (*Bonet-Ponce et al., 2020*; *Etoh and Fukuda, 2019*). Among the most interesting candidates linking BDNF-signalling and LRRK2 activity is Vps35, a component of the retromer complex that has been shown to modulate LRRK2 (*Mir et al., 2018*) and is recruited by TrkB binding to SorLA (*Rohe et al., 2013*), a sortilin family member. On the other hand, activation of Akt has been shown to compensate impairments of the insulin-stimulated GLUT4 trafficking associated with LRRK2 deficiency (*Funk et al., 2019*), opening the possibility that other downstream kinases could regulate Rab10 by LRRK2-independent mechanisms.

Once internalised, axonal TrkB reaches early endosomes, from where it can either recycle back to the plasma membrane, thus fine-tuning the response of nerve terminals to BDNF (reviewed in *Andreska*

*et al., 2020*) or undergo sorting to the retrograde axonal transport route, propagating neurotrophic signals to the soma (*Deinhardt et al., 2006*; *Ha et al., 2008*; *Zhou et al., 2012*). Given that Rab10 downregulation decreased retrograde transport of TrkB and stimulation with BDNF promoted antero-grade transport of Rab10, we reasoned that delivery of Rab10 to axon terminals facilitated the sorting of TrkB to retrogradely transported organelles. We demonstrated that Rab10DN expression increases the accumulation of internalised TrkB in early endosomes without significantly affecting recycling (*Figure 7*), suggesting that the pool of TrkB receptors available for local recycling and axonal transport are spatially segregated in the early endosome membrane, and thus indicating that functional Rab10 aids the exit of TrkB from early endosomes and its sorting to distinct axonal retrograde organelles.

When overexpressed, Rab10 is found at detectable levels in retrograde signalling endosomes posi-tive for TrkB or p75^NTR (*Figure 4*). An interesting observation is that only p75 receptors are present in Rab10-containing anterograde organelles, which bears the question on the role of the anterograde delivery of p75^NTR to distal axons. In sympathetic neurons, it has been shown that p75^NTR is rapidly mobilised to the plasma membrane upon stimulation with NGF via activation of Arf6 (*Hickman et al.,*

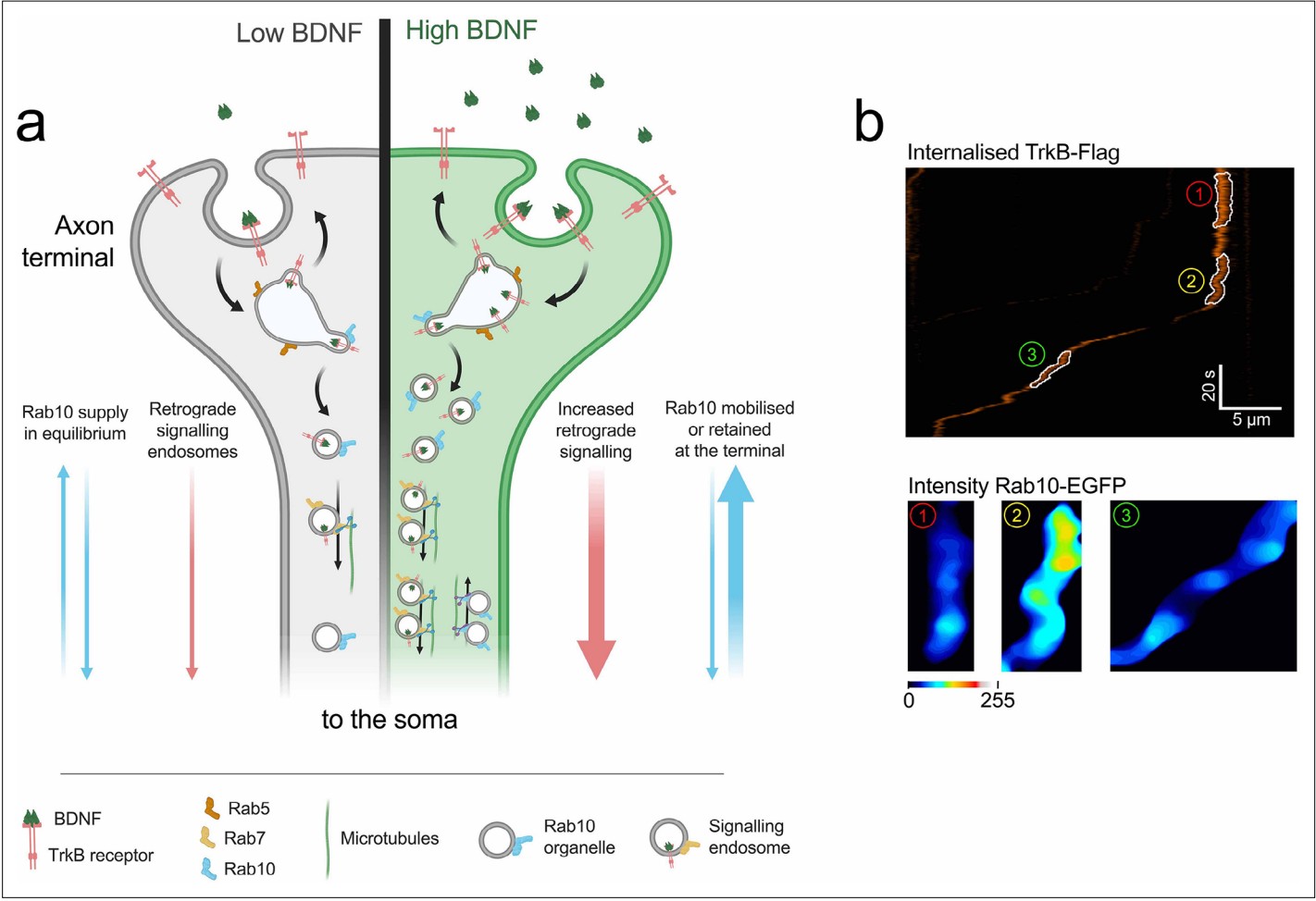

**Figure 8.** Model: role of Rab10 in the sorting of TrkB to retrograde axonal transport. (**a**) At steady state, low concentrations of BDNF (grey side of the terminal) induce basal levels of TrkB internalisation. Rab10 supply (blue arrows) is in equilibrium and it mediates a baseline level of TrkB retrograde transport (pink arrows). Upon increase of BDNF (green side of the axon terminal), TrkB endocytosis as well as the proportion of Rab10 organelles moving towards the axon terminal are increased. Increased amounts of Rab10 result in further facilitation of the sorting of TrkB out of early endosome and an augmented flow of retrograde signalling carriers. (**b**) This model predicts that the transition from stationary early endosomes to processive retrograde carriers would be preceded by the focal recruitment of Rab10. The top panel shows an example of a kymograph of internalised TrkB-Flag in the axon. In the kymograph, three segments have been highlighted: (1) stationary phase, (2) transition phase, and (3) retrograde transport phase. The bottom panel shows the levels of EGFP-Rab10 in these segments, which clearly demonstrates the enhanced recruitment of Rab10 during the transition phase, before onset of transport.

*2018*), a small GTPase that has been shown to share common organelles with Rab10 in *Caenorhabditis elegans* (*Shi and Grant, 2013*).

At endogenous levels of Rab10, however, when we analysed triple co-localisation of internalised $H_CT$ and TrkB with endogenous Rab10, we found that the absolute amount of retrograde TrkB associated to Rab10 organelles remains low (*Figure 3e*), suggesting that this interaction is transient. This is in line with previous evidence showing that in MDCK cells, Rab10 mediates transport from early endosomes to a polarised trafficking route (*Babbey et al., 2006*). Moreover, in *C. elegans*, Rab10 is recruited to early endosomes where it downregulates Rab5, helping to select cargoes for delivery to recycling endosomes (*Liu and Grant, 2015*). According to this model, the transfer of TrkB from a stationary early endosome to a retrograde carrier would be preceded by a local increase of Rab10. In *Figure 8b*, we show an example of a live-cell imaging experiment in which a TrkB-positive endosome makes the transition from stationary to retrograde. We show the level of EGFP-Rab10 during the stationary phase (red), in the transition phase immediately preceding movement (yellow), and during processive retrograde transport (green). Crucially, Rab10 is recruited to this TrkB-positive axonal organelle few seconds before it starts moving retrogradely, strengthening the validity of our working model (*Figure 8*).

Rab10 has been shown to participate in the biogenesis of tubular endosomes in mammalian cells (*Etoh and Fukuda, 2019*) and regulates endosomal phosphatidylinositol-4,5-bisphosphate in *C. elegans,* suggesting that this small GTPase modulates the membrane recruitment of factors altering membrane curvature and budding (*Shi and Grant, 2013*). During our live-cell imaging experiments, we observed Rab10-positive tubular organelles moving rapidly into proximal neurites, confirming that these structures are also generated in hippocampal neurons (*Figure 5—video 3*). In addition, it has been shown that snapin, an adaptor for cytoplasmic dynein recruitment to TrkB-signalling endosomes (*Zhou et al., 2012*), is also phosphorylated by LRRK2 (*Yun et al., 2013*), opening the possibility that Rab10 membranes constitute a specialised sorting domain. Interestingly, one of the well-characterised roles of KIF13B in the brain is the transport of phosphatidylinositol-3,4,5-biphosphate via its interactor centaurin-alpha (*Hammonds-Odie et al., 1996*), which is essential for axon specification (*Horiguchi et al., 2006*). Further research on the composition and cargo content of Rab10 organelles in neurons is therefore warranted.

Our data suggests that Rab10 regulates the amount of internalised TrkB that is sorted to retrograde signalling endosomes in response to the concentration of BDNF at axon terminals. Since BDNF is known to be released post-synaptically as a function of neuronal activity (*Matsuda et al., 2009*), retrograde neurotrophic signalling from the axon terminal is a substantial feedback mechanism regulating growth and survival, and therefore, ensuring that active circuits are preserved. To keep this feedback signal meaningful, any change in the availability of BDNF at the synapse must translate into proportional changes in the intensity of the signal arriving at the soma. We propose that Rab10-positive membranes deliver crucial components of the sorting machinery on demand. At steady state, the anterograde and retrograde flow of Rab10 compartments is in equilibrium (*Figure 8*; low BDNF). Further decrease in BDNF concentration makes retrograde transport of Rab10 carriers predominant, as observed in live-cell imaging experiments performed in neurons treated with anti-BDNF blocking antibodies (*Figure 4*; no BDNF). In contrast, adding BDNF reverts the directional bias to anterograde (*Figure 8*; high BDNF), which allows the delivery of appropriate levels of the sorting machinery to nerve terminals, thus increasing the efficiency of TrkB retrograde transport.

An independent example of anterograde delivery of components for the sorting of retrograde signalling molecules has been hypothesised for the bone-morphogenetic protein (BMP) pathway in *Drosophila* motor neurons. In this work, Khc-73, the fly orthologue of KIF13B, has been shown to regulate BMP sorting from early endosomes at the neuromuscular junction (*Liao et al., 2018*). This suggests that the proposed mechanism is evolutionary conserved in different neuronal types, increasing its potential as a therapeutic target for pathologies in which neurotrophic signalling from distal axons is impaired.

Little is known about the specific sorting machinery required for the biogenesis of signalling endosomes. Endophilins A1 and A3 have been shown to regulate the trafficking of TrkB across early endosomes and mediate survival signalling (*Burk et al., 2017*; *Fu et al., 2011*). Interestingly, endophilin A1 as well as Rab10 are known LRRK2 substrates, opening the possibility of both being found on the same organelle (*Matta et al., 2012*). Moreover, KIF13B has also been found to be enriched in early

endosomes (*Bentley et al., 2015*). The characterisation of the sorting machinery delivered to axonal terminals by Rab10-positive carriers will be crucial to understand not only how this mechanism allows coordination between local signalling and global neuronal responses, but also how this process may fail in neurodegeneration. In this light, promoting the delivery of Rab10 organelles to nerve terminals may be explored as an novel therapeutic strategy for diseases in which the endolysosomal system is overloaded or dysfunctional, such as Alzheimer's disease (*Van Acker et al., 2019*; *Xu et al., 2018*), or to increase the ability of axons to respond to trophic factors during regeneration.

# Materials and methods

## Key resources table

| Reagent type (species) or resource | Designation | Source or reference | Identifiers | Additional information |
|---|---|---|---|---|
| Antibody | Goat polyclonal anti-Rab10 | Santa Cruz Biotechnologies | Cat#sc-6564; RRID:AB_2237844 | 1:50 |
| Antibody | Mouse monoclonal anti-Rab10 | Abcam | Cat#ab104859; RRID:AB_10711207 | 1:200 |
| Antibody | Rabbit monoclonal anti-Rab10 | Cell Signalling | Cat#8127; RRID:AB_10828219 | 1:200 |
| Antibody | Rabbit monoclonal anti-TrkB | Merck (Millipore) | Cat#AB9872; RRID:AB_11214317 | 1:200 |
| Antibody | Chicken polyclonal anti-tubulin ßIII | Synaptic Systems | Cat#302 306; RRID:AB_2620048 | 1:300 |
| Antibody | Rabbit monoclonal anti-phosphorylated CREB | Abcam | Cat#ab32096; RRID:AB_731734 | 1:250 |
| Antibody | Mouse monoclonal anti-Rab7 | Abcam | Cat#ab50533, RRID:AB_882241 | 1:200 |
| Antibody | Rabbit polyclonal anti-Rab5 | Abcam | Cat#ab13253; RRID:AB_299796 | 1:200 |
| Antibody | Rabbit polyclonal anti-KIF13B | Bioss | Cat#bs-12387R; RRID:AB_2895287 | 1:200 |
| Antibody | Rabbit polyclonal anti-KIF5B | Abcam | Cat#ab5629; RRID:AB_2132379 | 1:200 |
| Antibody | Mouse monoclonal anti-GFP (B-2) | Santa Cruz Biotechnologies | Cat#sc-9996; RRID:AB_627695 | 1:1000 |
| Antibody | Mouse monoclonal anti-HA (12CA5) | Cancer Research UK | Cat#12CA5; RRID:AB_2920713 | 1:1000 |
| Antibody | Mouse monoclonal anti-Flag (M1) | Sigma | Cat#F3040; RRID:AB_439712 | 1:200 |
| Antibody | Rabbit polyclonal anti-p75[NTR] | Cancer Research UK | Cat#CRD5410; RRID:AB_2864325 | 1:200 |
| Recombinant DNA reagent | TET ON Advance | Takara Bio (Clontech) | Cat#630930 | |
| Recombinant DNA reagent | pLVX shRNA Rab10 | This study | | The shRNA MSH031352 from GeneCopoeia targeting *Rab10* has been cloned into a pLVX tight puro plasmid |
| Recombinant DNA reagent | pLVX mCherry-Rab10 | This study | | A mouse mCherry-Rab10 has been cloned into a pLVX tight puro inducible lentiviral vector using XbaI/NheI |
| Recombinant DNA reagent | pLVX myc-Rab10 (shRNA resistant) | This study | | From the pLVX mCherry-Rab10, mCherry has been replaced by myc, and 3 silent mutations have been introduced |
| Recombinant DNA reagent | pEGFP-C1 | Clontech | | Discontinued |

*Continued on next page*

*Continued*

| Reagent type (species) or resource | Designation | Source or reference | Identifiers | Additional information |
|---|---|---|---|---|
| Recombinant DNA reagent | EGFP-Rab10 WT | DOI:10.1111/j.1462–5822.2010.01468.x | RRID:Addgene_49472 | Marci Scidmore lab |
| Recombinant DNA reagent | EGFP-Rab10 T23N; Rab10DN | DOI:10.1111/j.1462–5822.2010.01468.x | RRID:Addgene_49545 | Marci Scidmore lab |
| Recombinant DNA reagent | EGFP-Rab10 Q68L | DOI:10.1111/j.1462–5822.2010.01468.x | RRID:Addgene_49544 | Marci Scidmore lab |
| Recombinant DNA reagent | HA-Rab10 WT | MRC Protein Phosphorylation and Ubiquitylation Unit | DU44250 | Dario Alessi lab |
| Recombinant DNA reagent | GFP-KIF13B | 10.1111/tra.12692 | RRID:Addgene_134626 | Marvin Bentley lab |
| Recombinant DNA reagent | TrkB-FLAG | 10.1091/mbc.e05-07-0651 | | Francis Lee lab |

## Neuronal cultures

Embryonic hippocampal neurons from C57BL/6 mice of either sex and embryonic age of 16–17 days were dissected adapting previously described protocols (*Kaech and Banker, 2006*). Dissection was performed in cold Hanks' balanced salt solution (HBSS) and the tissue was collected in cold Hibernate E medium (Thermo Fisher, #A1247601). After incubating for 10 min in 300 µL of Accumax (Innovative Cell Technologies, #AM105) diluted in HBSS (1:1), tissue was washed in HBSS, resuspended in warm plating medium (Minimum Essential Medium supplemented with 10% horse serum, 0.6% glucose, and 2 mM glutamine), and mechanically dissociated by pipetting. 10,000–12,000 neurons per $cm^2$ were then seeded on glass coverslips or microfluidic chambers, pre-coated with 1 mg/mL poly-L-lysine. Before coating, glass coverslips were treated overnight with NoChromix (Godax Laboratories), washed three times and sterilised in 70% ethanol. Microfluidic chambers were produced in-house as previously described (*Restani et al., 2012*; *Sannerud et al., 2011*). Polydimethylsiloxane inserts were fabricated from resin moulds, which are replicas of the master template produced by soft lithography, and then irreversibly bound to glass-bottom dishes (WillCo Wells, #HBSB-3512) by plasma treatment. Neurons were left in plating medium for 1.5 hr and then shifted to maintenance medium (Neurobasal supplemented with B27, 2 mM glutamine, 0.6% glucose, and antibiotics). Half of the culture medium was replaced by fresh medium every 3–4 days.

## Immunofluorescence

Cells were washed in phosphate buffer saline (PBS) and fixed for 15 min in 3% paraformaldehyde (PFA) and 4% sucrose dissolved in PBS. Next, they were incubated in 0.15 M glycine dissolved in PBS for 10 min and then blocked and permeabilised simultaneously by incubation in 5% bovine serum albumin (BSA) and 0.1% saponin in PBS for 1 hr. Samples were incubated at 4°C overnight with primary antibodies diluted in 5% BSA, 0.05% saponin, 0.1 mM $CaCl_2$, and 0.1 mM $MgCl_2$ dissolved in PBS at the concentrations indicated in the Key resources table. Then, cells were washed three times with PBS and incubated for 90 min with Alexa Fluor-conjugated secondary antibodies 1:400 (Thermo Fisher) in 5% BSA, 0.05% saponin, 0.1 mM $CaCl_2$, and 0.1 mM $MgCl_2$ dissolved in PBS. 4',6-diamidino-2-phenylindole (DAPI) was added with the secondary antibodies when appropriate. Finally, coverslips were washed in PBS and mounted with Mowiol.

## Rab10 knockdown

Rab10 was knocked down by transducing 5 days in vitro (DIV) hippocampal neurons with an inducible shRNA *Rab10* lentivirus (pTightPuro-shRNA Rab10) and its doxycycline-dependent regulator TET-ON Advance (Clontech). After 48 hr, they were treated with doxycycline 1 µg/mL and kept in the incubator for 12, 18, 24, 36, and 48 hr. Cell density, levels of Rab10 and TrkB, as well as the general health of

the culture were analysed at 12, 24, and 48 hr to establish the optimal time frame for the following experiments.

## Transfection and plasmids

Hippocampal neurons were transfected at 7 DIV using Lipofectamine 2000 (Thermo Fisher, Cat# 11668019). Experiments were carried out after 20–24 hr. The pEGFP-C1 plasmid is from Clontech (Addgene plasmid # 13031, RRID:Addgene_13031), the plasmids for EGFP-Rab10 WT (RRID:Addgene_49472), EGFP-Rab10 T23N (RRID:Addgene_49545), and EGFP-Rab10 Q68L (RRID:Addgene_49544) were a gift from Marci Scidmore (*Huang et al., 2010*), TrkB-FLAG plasmid was a gift from Francis Lee (*Chen et al., 2005*). Neuro-2a cells were transfected using Lipofectamine 3000 (Thermo Fisher, Cat# L3000001) and the experiments were done 48 hr later. The GFP-KIF13B plasmid (RRID:Addgene_134626) was a gift from Marvin Bentley (*Yang et al., 2019*), whilst the HA-Rab10 plasmid was provided by Dario Alessi and Miratul Muqit (Dundee University, DU44250).

## Retrograde accumulation and signalling assays

Hippocampal neurons were seeded in custom-made microfluidic devices and after 5 DIV they were transduced with the inducible shRNA *Rab10* and TET-ON Advance lentiviruses. At 7 DIV, dishes with overt axon crossing were selected and doxycycline 1 µg/mL was added to the cell bodies, 18–22 hr later the media was replaced with Neurobasal in somatic and axonal compartments to deplete cells from endogenous growth factors for 1 hr. For analysing the retrograde accumulation of TrkB, we added polyclonal antibodies against the extracellular domain of TrkB (1:50 rabbit anti-TrkB, Millipore, Cat# AB9872, RRID:AB_2236301) together with 20 ng/mL BDNF for 2.5 hr. After PFA fixation, the transport of internalised antibodies was revealed by incubating the somatic compartment with fluorescently labelled secondary antibodies. The same protocol was used to study retrograde propagation of neurotrophic signalling; after 1 hr of growth factor depletion, axons were stimulated with 20 ng/mL BDNF for 2.5 hr, and after fixation, phosphorylation of CREB in the nucleus was analysed by immunofluorescence. Transduction with a myc-Rab10 containing three silent mutations was used for rescue.

## Co-localisation studies

To analyse the presence of two markers in the same organelle, we used confocal z-stack images (voxel size: 0.197 × 0.197 × 0.5 µm) and the confined displacement algorithm to measure Manders' correlation index within axons and determine its statistical significance compared to random images of identical total intensity and shape (*Ramírez et al., 2010*). To compute random scenarios, seven random radial displacements were taken at a maximum radial distance of 12 pixels (a total of 353 samples), and histograms binning = 16. CDA was implemented by using the plugin from the GDSC University of Sussex (http://www.sussex.ac.uk/gdsc/intranet/microscopy/UserSupport/AnalysisProtocol/imagej/colocalisation). All the data points were plotted and the mean and standard error are indicated for each group and compared using Student's *t*-test. Statistical significance of the individual data point is colour-coded (see figure legends).

## Super-resolution radial fluctuations

High-fidelity super-resolution information was extracted from time series of 1000 confocal images per channel by using super-resolution radial fluctuations (SRRF) algorithm (*Culley et al., 2018b*). Super-resolution images were then quality-controlled by using Super-Resolution Quantitative Image Rating and Reporting of Error Locations (SQUIRREL) algorithm (*Culley et al., 2018a*). Implementation of the algorithms was done in FIJI by using the open-source plugin NanoJ-core (https://henriqueslab.github.io/resources/NanoJ).

## Immunoendocytosis

Hippocampal neurons transfected with TrkB-Flag were kept in Neurobasal media for 1 hr and then incubated on ice with 1:50 mouse anti-Flag antibody (Sigma-Aldrich, Cat# F3040, RRID:AB_439712). In selected samples, 20 nM $H_C$T was also added (*Deinhardt et al., 2006*). Internalisation of receptors was then induced by incubation with 50 ng/mL BDNF for 30 min at 37°C. Antibodies bound to receptors still at the cell surface were dissociated by washing twice for 1 min in PBS supplemented

with 1 mM EDTA. In experiments to measure recycling of internalised receptors, neurons were further stimulated with BDNF for other 60 min in the presence of an Alexa Fluor647-conjugated anti-mouse secondary antibody, fixed, permeabilised, and incubated with an Alexa Fluor555-conjugated anti-mouse secondary antibody to detect total internalised receptor.

## Axonal transport

Axonal transport of overexpressed fluorescent proteins and internalised fluorescent antibodies was analysed from confocal time series of 1 frame/s and a pixel size of ~0.1 × 0.1 μm$^2$, captured during 5 min intervals at different time points by using a Zeiss LSM 780 NLO multiphoton confocal microscope with an oil immersion ×63 objective and equipped with an environmental chamber (Zeiss XL multi S1 DARK LS set at 37°C and environmental $CO_2$). For these experiments, neurons were cultured on 25 mm coverslips kept in Neurobasal for 1 hr prior to cell imaging and mounted inside Attofluor chambers (Thermo Fisher Scientific, Cat# A7816) with BrightCell NEUMO photostable media (Sigma-Aldrich, Cat# SCM145) supplemented with 10 mM HEPES. Speed, pausing, and direction of labelled organelles were analysed from kymographs by using Kymoanalyzer set of macros from Encalada lab (https://www.encalada.scripps.edu/kymoanalyzer; *Neumann et al., 2017*).

## Co-immunoprecipitation

Neuro-2A cells have been originally obtained from ATCC and modified to stably express TrkB-Flag (*Terenzio et al., 2014*). Their identity has been confirmed in the future batches by assesing their morphology and probing for TrkB-flag. They have been tested for mycoplasma contamination routinely. Cells transfected with plasmids encoding HA-Rab10 and GFP-KIF13B were starved for 1 hr, stimulated or not with BDNF for 30 min at 37°C, and then scrapped and incubated at 4°C for 30 min in lysis buffer containing 25 mM 4-morpholine-propanesulfonic acid sodium salt (MOPS) pH 7.2, 100 mM KCl, 10 mM $MgCl_2$, 1% octyl-phenoxy-polyethoxyethanol (IGEPAL CA-630, Sigma-Aldrich, Cat# I3021) and HALT proteases and phosphatases inhibitor (Thermo Fisher, Cat# 78425). After clearing by centrifugation at 21,000 × $g$ for 10 min, the detergent concentration was adjusted to 0.5%. Pre-washed anti-HA magnetic beads (Pierce, Cat# 88837) were incubated with lysates overnight at 4°C. Samples were washed in lysis buffer five times and elutes in loading buffer (NuPAGE LDS sample buffer, Thermo Fisher, Cat# NP0007, supplemented with 50 mM DTT) at 95°C for 10 min. The levels of immunoprecipitated Rab10 and KIF13B were analysed by Western blot using mouse anti-HA (Cancer Research UK, Cat# 12CA5, RRID:AB_2920713) and mouse anti-GFP B-2 (Santa Cruz Biotechnologies, Cat# sc-9996, RRID:AB_627695) antibodies.

## Statistical analysis

Data generated in independent experiments were tested for normality and homoscedasticity to apply the appropriate corrections to the statistical tests. Each specific test, its degrees of freedom, and level of significance are indicated in the respective figure legends. Further details are summarised in the statistical annex (*Table 1*). Plots show mean ± standard error, and exact p-values are indicated when relevant.

## Software

Images were handled, edited, and analysed using ImageJ/FIJI (version 2.1.0, 1.53c). Figures were checked with Coblis (https://www.color-blindness.com/coblis-color-blindness-simulator/) using The Colour Blind Simulator algorithms from Matthew Wickline and the Human-Computer Interaction Resource Network. Palettes were adjusted to maximise visibility. The Orange/Green/Purple balanced look-up table was obtained from Christophe Leterrier's GitHub repository (https://github.com/cleterrier/ChrisLUTs, copy archived at *Lazo, 2023a*). Data were imported, analysed, and sorted as R files using RStudio (version 1.0.44). GraphPad Prism for Mac (version 6.00, GraphPad Software) was used for running statistical analysis and generate the plots included in the figures. Illustrations were created with BioRender (http://www.biorender.com). Updated versions of ImageJ macros and R scripts used in this article, as well as the specific implementation of Kymoanalyzer used to analyse our datasets, can be found on our GitHub repository (https://github.com/omlazo; copies archived at *Lazo, 2023b*, *Lazo, 2023c* and *Lazo, 2023d*).

**Table 1.** Statistical summary.

| Figure | | Variable | Test | Groups | Degrees of freedom | p value |
|---|---|---|---|---|---|---|
| | | | Two-way ANOVA | Knock down; time; interaction | F(1,68) | 0.0270; 0.0406; 0.2114 |
| | | | Multiple comparisons | Control: shRNA Rab10 at 24 hr | t(68) | >0.9999 |
| | b | Cells per field | Multiple comparisons | Control: shRNA Rab10 at 48 hr | t(68) | 0.0183 |
| | | | Two-way ANOVA | Knock down; time; interaction | F(1,653); F(2,653); F(2,653) | <0.0001 |
| | | | Multiple comparisons | Control: shRNA Rab10 at 12 hr | t(653) | 0.2012 |
| | | | Multiple comparisons | Control: shRNA Rab10 at 24 hr | t(653) | <0.0001 |
| Figure 1 | c | Rab10 expression | Multiple comparisons | Control: shRNA Rab10 at 48 hr | t(653) | <0.0001 |
| | d | TrkB accumulation | Unpaired Student's t | Control: shRNA Rab10 | t(140) | <0.0001 |
| | e | TrkB accumulation and Rab10 expression | Pearson r | Control: shRNA Rab10 | XY pairs = 131 | <0.0001 |
| | | | One-way ANOVA | Control, shRNA Rab10 and rescue | F(2,280) | <0.0001 |
| | | | Multiple comparisons | Control: shRNA Rab10 | t(280) | <0.0001 |
| Figure 2 | g | pCREB abundance | Multiple comparisons | Control: rescue | t(280) | 0.0336 |
| | c | Co-localisation Rab10 and Rab5 | Unpaired Student's t | M1 control: BDNF | t(46.27) | 0.0244 |
| | | | Unpaired Student's t | M2 control: BDNF | t(48.94) | 0.4794 |
| | d | Co-localisation Rab10 and Rab7 | Unpaired Student's t | M1 control: BDNF | t(72.32) | 0.0621 |
| | | | Unpaired Student's t | M2 control: BDNF | t(62.33) | 0.1043 |
| | | Area of overlay HcT and TrkB (retrograde TrkB) | One-way ANOVA | 30, 60, and 90 min | F(2,72) | <0.0001 |
| | | | Multiple comparisons | 30:60 min | t(72) | >0.9999 |
| | | | Multiple comparisons | 60:90 min | t(72) | 0.0002 |
| | | | One-way ANOVA | 30, 60, and 90 min | F(2,72) | 0.2730 |
| | | | Multiple comparisons | 30:60 min | t(72) | 0.5717 |
| Figure 3 | e | Area of overlay retrograde TrkB and Rab10 | Multiple comparisons | 60:90 min | t(72) | >0.9999 |
| | | | Unpaired Student's t | Anterograde pre: post BDNF | t(14) | 0.0150 |
| | | | Unpaired Student's t | Retrograde pre: post BDNF | t(14) | 0.0030 |
| Figure 5 | c | Direction of Rab10 organelles | Unpaired Student's t | Non-mobile pre: post BDNF | t(14) | 0.4278 |

*Table 1 continued on next page*

*Table 1 continued*

| Figure | | Variable | Test | Groups | Degrees of freedom | p value |
|---|---|---|---|---|---|---|
| | b | KIF13B intensity ratio | Kolmogorov–Smirnov Student's *t* | Control: BDNF | t(79.5) | 0.0177 |
| | c | KIF13B area occupancy ratio | Kolmogorov–Smirnov Student's *t* | Control: BDNF | t(79.5) | 0.8024 |
| | e | KIF5B intensity ratio | Kolmogorov–Smirnov Student's *t* | Control: BDNF | t(62.5) | 0.0644 |
| | f | KIF5B area occupancy ratio | Kolmogorov–Smirnov Student's *t* | Control: BDNF | t(62.5) | 0.3738 |
| *Figure 6* | h | KIF13B co-immunoprecipitation | Paired Student's *t* | Control: BDNF | t(6) | 0.0024 |
| | c | Recycling of TrkB | Unpaired Student's *t* | EGFP: Rab10 DN | t(60.40) | 0.3914 |
| *Figure 7* | e | Co-localisation TrkB and Rab5 | Unpaired Student's *t* | EGFP: Rab10 DN | t(38.22) | 0.0080 |

## Acknowledgements

We thank Dr James N Sleigh (UCL Queen Square Institute of Neurology) for critical reading of the first manuscript. This work was supported by the MRC Project grant MR/T001976/1 [OML], the Wellcome Trust Senior Investigator Awards (107116/Z/15/Z and 223022/Z/21/Z) [GS] and the UK Dementia Research Institute Foundation award UKDRI-1005 [GS].

## Additional information

### Funding

| Funder | Grant reference number | Author |
|---|---|---|
| Medical Research Council | MR/T001976/1 | Oscar Marcelo Lazo |
| Wellcome Trust | 107116/Z/15/Z | Giampietro Schiavo |
| Wellcome Trust | 223022/Z/21/Z | Giampietro Schiavo |
| UK Dementia Research Institute Foundation | UKDRI-1005 | Giampietro Schiavo |

The funders had no role in study design, data collection and interpretation, or the decision to submit the work for publication. For the purpose of Open Access, the authors have applied a CC BY public copyright license to any Author Accepted Manuscript version arising from this submission.

### Author contributions

Oscar Marcelo Lazo, Conceptualization, Formal analysis, Funding acquisition, Investigation, Visualization, Methodology, Writing – original draft, Project administration, Writing – review and editing; Giampietro Schiavo, Conceptualization, Resources, Supervision, Funding acquisition, Project administration, Writing – review and editing

### Author ORCIDs

Oscar Marcelo Lazo http://orcid.org/0000-0003-4542-482X
Giampietro Schiavo http://orcid.org/0000-0002-4319-8745

### Ethics

Animal experimentation: The use of mice during this study is regulated by the Home Office Project Licence PCF5CF564 granted to Prof. Schiavo on 19.12.2018. The number of animals used and the protocols for their handling minimising stress and suffering, are approved and monitored by University College London Biological Services. No human samples or personal data will be collected during

this study. Procedures and research practices are in agreement with the UCL Code of Conduct for Research and Singapore Statement on Research Integrity (2010).

### Decision letter and Author response
Decision letter https://doi.org/10.7554/eLife.81532.sa1
Author response https://doi.org/10.7554/eLife.81532.sa2

## Additional files

### Supplementary files
• Transparent reporting form

### Data availability
Source tables for all the data in the manuscript have been submitted as a supplementary file (source data files 1-7).

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
