## [Editor Report]

This important study, of interest to cellular neurobiologists, uses convincing microscopy methods to show that Rab10 GTPase is a new regulator of neurotrophin receptor trafficking and signaling. Defining how neurons respond to spatial extrinsic cues, such as neurotrophins, and relay this information long-distance to influence transcriptional events is an important topic in neurobiology.

---

## [Decision Letter]

**Decision letter after peer review:**

[Editors’ note: the authors submitted for reconsideration following the decision after peer review. What follows is the decision letter after the first round of review.]

Thank you for submitting the paper "Rab10 regulates the sorting of internalised TrkB for retrograde axonal transport" for consideration by *eLife*. Your article has been reviewed by 2 peer reviewers, and the evaluation has been overseen by a Senior Editor. The reviewers have opted to remain anonymous.

Comments to the Authors:

We are sorry to say that, after consultation with the reviewers, we have decided that this work will not be considered further for publication by *eLife* in its current form, but we would happily reconsider a new submission that addresses the reviewers' concerns in full.

As described below, the reviewers felt that the story, while potentially very interesting, was too preliminary in its current form; their specific concerns are described below.

*Reviewer #1 (Recommendations for the authors):*

Here, the authors use confocal microscopy, super resolution, and live imaging to report a role for Rab10 in the sorting of internalized TrkB receptors and in propagating the BDNF signal from axon terminals to the soma. The imaging analyses, specifically the super-resolution and live imaging, are well-done. However, the manuscript appears to be preliminary at this point; the authors draw strong conclusions about the mechanism of action of Rab10 in TrkB sorting that remain to be fully supported.

Concerns:

1. The authors see little co-localization between endogenous Rab10 and TrkB. The conclusion that Rab10 and TrkB are retrogradely co-transported is based on Rab10 over-expression, which raises concerns about the specificity of the interactions and artifacts related to over-expressing a Rab GTPase.

2. Other than P-CREB immunofluorescence experiments, the authors do not provide data to show the functional effects of Rab10 depletion on BDNF-mediated trophic effects. The authors state in the Discussion that Rab10 knockdown perturbs neuronal morphogenesis and survival, but do not show this data. These data are necessary to strengthen the conclusion that Rab10 is necessary to mediate BDNF-dependent trophic signaling. On a related note, the color intensity scales to show TrkB or P-CREB levels in cell bodies with Rab10 depletion are not adequate.

3. It is interesting that BDNF enhances anterograde Rab10 transport based on live imaging. Additional analyses are needed to strengthen the conclusion that Rab10 is enriched in distal axons and to define the mechanisms (phosphorylation?) underlying Rab10 recruitment/retention

4. The anterograde motor Kif13B is suggested to be involved in BDNF-induced anterograde transport of Rab10 based on confocal microscopy analyses in fixed neurons. However, this aspect is not fully developed. The authors should determine if Kif13B directly interacts with Rab10 and the necessity of Kif13B for anterograde Rab10 transport, retrograde transport of TrkB receptors or BDNF-mediated trophic effects.

5. The suggestion that Rab10 regulates "sorting" of internalized TrkB receptors is very interesting. The authors show nicely (in Figure 6) that dominant negative Rab10 (rab10DN) does not disrupt TrkB recycling in axons, but increases the co-localization of TrkB with Rab5. The images in Figure 6d also show that Rab5 domains appear to be larger in the Rab10 DN expressing neurons, which is consistent with early endosomes being perturbed. However, additional analyses are needed to define how exactly Rab10 coordinates with Rab5 and Rab7 in "handing off" TrkB from early endosomes to late endosomes in axons. Which axonal segments were imaged in the analyses in Figure 6? Was this in the distal axons? What happens to TrkB flux in mid-axons and retrograde accumulation in cell bodies in the Rab10 DN or shRNA expressing neurons? Does expression of Rab10 shRNA or Rab10 DN decrease co-localization of retrogradely transported TrkB with Rab7?

6. The authors should include important controls to show the specificity of Rab antibodies, and the knockdown of Rab10 using shRNA. In Figure 1f, the Rab10 levels (assessed by Rab10 immunofluorescence) appear to not be significantly affected by the kd or rescue, which raises questions about the antibody specificity, as well as efficacy of the kd.

Comments above address concerns about conclusions that are not fully supported by the data. Here are some specific suggestions for additional analyses plus other recommendations.

1. The authors should use additional measures to support that Rab10 associates/co-migrates with TrkB, for example by biochemical methods, or by performing live imaging of co-transport in neurons transfected with Rab10-GFP but with endogenous Rab10 knocked down to mitigate the OE concern.

2. The authors should strengthen the data to support that Rab10 accumulates in distal axons in a BDNF-dependent manner, using confocal microscopy or biochemical means. Live imaging analyses can be used to determine if Rab10 is being mobilized long-distance from cell bodies by the retrograde BDNF signal, or if this is a local effect of BDNF on Rab10 recruitment to the axon terminals.

3. In Figure 1c, the authors show reduced accumulation of axon-derived TrkB receptors in neurons expressing Rab10 shRNA. Images of the axons should be included to assess whether the TrkB receptors accumulate in distal axons, as proposed in the model (Figure 7)

4. The authors also propose a mechanism for BDNF-induced enrichment of Rab10 in axons that involves Rab10 phosphorylation. The manuscript would be strengthened by additional analyses to define mechanisms underlying Rab10 recruitment. The imaging analyses of Kif13B could be complemented with biochemical methods to show BDNF-induced association with Rab10.

5. Rab10 has been proposed to play a role in generating tubular endosomes-it would be of interest to use super-resolution imaging to define the morphology of TrkB/Rab10-positive endosomes in distal axons.

6. The authors should assess whether the Rab10 kd or expression of Rab10 DN has general effects on neuronal viability or affect total TrkB levels, and not specifically on retrograde transport of axonally-derived TrkB receptors.

*Reviewer #2 (Recommendations for the authors):*

Signaling from neurotrophins requires endocytosis of the ligand-bound receptors (such as NGF-TrkA or BDNF-TrkB), sorting into signaling endosomes and subsequent transport retrogradely along the axon back to the soma for signaling. Multiple Rab proteins have been implicated in this trafficking pathway, most prominently Rab5 and Rab7. This manuscript shows that Rab10 is also required for retrograde arrival of signaling endosomes in the soma and subsequent signaling. The most interesting finding is that Rab10-positive organelles show increased anterograde transport in response to BDNF and are not part of the TrkB-positive retrogradely moving signaling endosome itself. Interference with Rab10 traps TrkB in Rab5-positive early endosomes, and the authors propose that Rab10 is required for sorting of TrkB into signaling endosomes. Since Rab10 is largely in a distinct set of axonal organelles, the point of conversion of endocytosed TrkB and Rab10 is not clear. The paper stops short of decisively answering where Rab10 is active to promote sorting and what BDNF signaling does to change Rab10 motility patterns.

1) Rab10 was enriched in SEs by SILAC/mass spec (from a previous paper by this group) but now is not in the retrograde carriers. Can you discuss?

2) Are the anterograde Rab10 carriers endosomally derived? It would be important to characterize the cargos of these carriers. I think the observation that BDNF affects the directionality of the Rab10 carriers is very intriguing. Were these in microfluidic chambers? How is BDNF signaling conveyed to Rab10 carriers?

2) There is a lot of conjecture about Rab10 vesicles delivering machinery to the distal axon early endosome, but no experiment to address this. Where is TrkB in Rab10 interference conditions? It seems to be stuck with Rab5 and not sorting back to the axonal plasma membrane or into retrograde carriers. Is it accumulating in distal axon tips? It would be good to have live imaging of TrkB in Rab10-DN conditions along axons. The recycling experiments (Figure 6) are very interesting, but it looks like this is happening along the axon shaft. Is this correct?

3) It is not always clear what experiments are done in microfluidic chambers. Please specify in each figure.

4) Is p75 not in a complex with TrkB after BDNF binding? What is the interpretation/implication of p75 being in different carriers in the axon? I find this a very interesting observation, but the relevance is not explained or further explored.

5) The data with kinesins is not adding much to the understanding of the pathway. No interference with any kinesin is performed.

3) All the discussion of LRRK2 and phosphorylation is besides the point since the authors do not test involvement of this regulatory mechanism. This would be very interesting, but as is, the paper stops short of decisively answering where Rab10 is active to promote sorting and what BDNF signaling does to change Rab10 motility patterns.

There are several very interesting observations, but the scope of the study is somewhat limited.

[Editors’ note: further revisions were suggested prior to acceptance, as described below.]

Thank you for resubmitting your work entitled "Rab10 regulates the sorting of internalised TrkB for retrograde axonal transport" for further consideration by *eLife*. Your revised article has been evaluated by Suzanne Pfeffer (Senior Editor) and a Reviewing Editor.

The manuscript has been improved but there are some remaining issues that need to be addressed, as outlined below:

The reviewers have discussed their reviews with one another, and the Reviewing Editor has drafted this to help you prepare a revised submission. We realize that this manuscript was already reviewed and resubmitted, and the reviewers hesitated to ask for more data, but they felt strongly that some additional work would be required to support the conclusions of your study. Note that it is not always possible to secure the same reviewers with each round of submission.

Essential revisions:

(The other issues that follow below are worthy of text clarification)

1) The fact that there is little colocalization between Rab10 and TrkB needs further characterization. That is not consistent with "a new Rab10 organelle" that mediates TrkB sorting and retrograde transport. Please test other recycling endosome Rabs, such as Rab11, Rab14, Rab10 to determine if these are really novel organelles or simply Rab11-recycling endosomes that also contain Rab10.

2) Please test whether Kif13B or Kif5B knock-down affects TrkB transport.

3) The IP experiment shown in Figure 6G-H is not very convincing. Can you blot for endogenous Kif13B co-precipitation with HA-Rab10? Second, it appears that there is more HA-Rab10 precipitated in BDNF-treated samples, thus, it is unclear whether slight increase in GFP-Kif13B co-precipitation is due to that rather than increased interaction with Rab10. Third, why there is some GFP-Kif13B in sample that does not express HA-Rab10? Please address.

*Reviewer #1 (Recommendations for the authors):*

In the revised manuscript, the authors faithfully addressed most of the reviewers' concerns. I have only few suggestions (see below).

1. Description of "Rab10-EGFP" and "Rab10-HA" should be considered, because EGFP and HA were tagged to the N-terminus of Rab10. Indeed, the authors have used "GFP-Rab5" in their previous paper (see ref. 12).

2. Several resources were missing in the Key resources table, e.g., shRNA-resistant Rab10, Rab10-Q68L, and JIP3/4 antibodies.

3. (Line 564) The reference may be mis-cited. Ref. 22 is correct?

*Reviewer #2 (Recommendations for the authors):*

This is a manuscript that focuses on understanding the machinery governing generation and transport of TrkB-containing endosomes, especially the involvement of Rab10 in the process. Since we are only beginning to understand the long-range signaling by endosomes in neurons, the topic is quite interesting and potentially significant. Additional data are needed to confirm association of endogenous Rab10 with TrkB-endosomes and the potential novelty of these structures.

1) Rab10 knock-down should be evaluated using western blotting or qPCR. It is much more quantitive method than microscopy. Additionally, how authors control for possible off-target effects? They used rescue in Figure 2f-g. Similar rescues should be used in other analyses in Figures 1-2.

2) Rescue images in Figure 2f are not particularly convincing. Not sure how authors can tell apart the neurons that are presumably expressing rescue plasmid.

3) As other reviewers pointed out there is very little colocalization between Rab10 and TrkB. That is not consistent with authors conclusions that Rab10 marks a new type of organelle that mediates TrkB sorting and retrograde transport. Authors speculate that these are transient associating but provide little evidence to support that. Overexpression of both, Rab10 and TrkB is hardly a strong evidence for that since even under these high overexpressed conditions the colocalization is still limited. Did authors try to overexpress (with TrkB) other recycling edosome Rabs, such as Rab11, Rab14, Rab10. Finally, Rab10 is related to Rab11, which is a well-established regulator of endocytic sorting and recycling. Does Rab10 colocalize with Rab11 (I suspect it will)? Are these really novel organelles or they are simply Rab11-recycling endosomes that also contains Rab10 (as it was shown in other experimental systems).

4) It is not clear why authors picked to analyze Kif5b and Kif13B. There are several other kinesins that were implicated in axonal transport. Why authors chose not to study them?

5) The change in Kif13B and Kif5B intensity in response to BDNF is very moderate at best. It does not help that authors do not show validation of anti-Kif5B or anti-Kif13B antibodies. It would also be good to test whether Kif13B or Kif5B knock-down affects TrkB transport.

6) IP experiment shown in Figure 6G-H is not very convincing. First, since authors used anti-Kif13B antibodies for immunofluorescent microscopy, why they did not blot for endogenous Kif13B co-precipitation with HA-Rab10. Second, it appears that there is more HA-Rab10 precipitated in BDNF-treated samples, thus, it is unclear whether slight increase in GFP-Kif13B co-precipitation is due to that rather than increased interaction with Rab10. Third, why there is some GFP-Kif13B in sample that does not express HA-Rab10?

7) Figure 7. Dominant-negative Rab mutants causes numerous non-specific effects. Since authors has Rab10 shRNA, all recycling experiments need to be done in Rab10-KD cells. Additionally, it is very puzzling that trapping TrkB in early endosomes did not affect recycling. Most plasma membrane receptors recycle by sequential transport from early endosomes to recycling endosomes. Consequently, I would expect that trappin TrkB in early endosomes would decrease its recycling.

*Reviewer #3 (Recommendations for the authors):*

Points #3, 5 and 6 of reviewer 2 seem very important.

The functional mechanism of Rab10 in BDNF signaling still remains a bit fuzzy since the mobilization of Rab10 in the axon terminal in response to BDNF is counterintuitive to explain the increased signaling to the soma, at least in the absence of further mechanistic insight. The model in Figure 8a thus remains speculative.

---

## [Author Response]

[Editors’ note: the authors resubmitted a revised version of the paper for consideration. What follows is the authors’ response to the first round of review.]

We want to thank you and the Reviewers for their positive comments and very insightful suggestions with regard to our manuscript titled “Rab10 regulates the sorting of internalised TrkB for retrograde axonal transport”. Their advice has greatly helped us to clarify the key messages of our manuscript and prompted us to perform additional experiments to make our work more impactful and interesting for the readership of *eLife*.

We appreciate your kind words while summarising the importance of our work: “defining how neurons respond to spatial extrinsic cues, such as neurotrophins, and relay this information longdistance to influence transcriptional events is an important topic in neurobiology”. This showed us that the main message of our work has been well recognised and valued. Moreover, the Reviewers explicitly mentioned what for us is the main focus of this work: “to report a role for Rab10 (…) in propagating the BDNF signal from axon terminals to the soma” and “that Rab10 is also required for retrograde arrival of signalling endosomes in the soma and subsequent signalling”. Reviewer #2 adds that “the most interesting finding is that Rab10-positive organelles show increased anterograde transport in response to BDNF and are not part of the TrkB-positive retrogradely moving signalling endosome itself”, whilst Reviewer #1 recognises the quality of “the imaging analyses, specifically the super-resolution and live imaging”.

However, there is also consensus regarding the need of a better characterisation of the specific role of Rab10 in the system and the mechanism involving KIF13B in the anterograde transport of Rab10 organelles. In this amended version of the manuscript, we have added a new figure documenting the efficacy of the Rab10 knock down, the effects of its depletion in cell survival and TrkB expression (Figure 1). Moreover, we have expanded our analysis of the BDNFdependent interaction between Rab10 and KIF13B (Figure 6g-h), as well as provided supplementary videos and representative data to support our observations. This new evidence has been carefully calibrated to keep in focus what the Reviewers have also recognised as the key message of our paper: the novel role of BDNF-regulated axonal transport of Rab10 organelles to propagate neurotrophic signalling from the axon to the nucleus of central neurons.

We address below the specific points raised by the Reviewers:

1. Reviewer #1 points out that while we see ”little co-localization between endogenous Rab10 and TrkB (…), the conclusion that Rab10 and TrkB are retrogradely co-transported is based on Rab10 over-expression, which raises concerns about the specificity of the interactions and artifacts related to over-expressing a Rab GTPase”. We agree with the reviewer regarding the fact that endogenous TrkB and Rab10 are unlikely to be retrogradely co-transported in the same organelle, as they correctly conclude from Figure 3e-f. Our data suggest that there is a transient association between Rab10 and the endosome containing TrkB. Therefore, we overexpressed wild-type Rab10 to create an experimental situation in which this transient association is stabilised long enough to be observed as events of co-transport. The reviewer suggested that we should “use additional measures to support that Rab10 associates/co-migrates with TrkB, for example by biochemical methods, or by performing live imaging of co-transport in neurons transfected with Rab10-GFP but with endogenous Rab10 knocked down to mitigate the OE concern”. As mentioned in our manuscript, we obtained independent evidence of the association of Rab10 with neurotrophin signalling endosomes in our early work describing a quantitative proteomic analysis of these organelles using an affinity purification approach in mouse motor neurons (see reference 18). Consistently with the conclusions of this work, we found that Rab10 is not accumulated on the surface of these organelles during signalling endosome maturation, as found for Rab7, further supporting the view that Rab10 interacts transiently with these organelles (see also our response to Reviewer #2 point 7 below). Furthermore, transfection of Rab10-EGFP, even when endogenous Rab10 is knocked down, results in eight fold increase on its expression levels; additionally, biochemistry may provide inconclusive results, as it will fail to reveal dynamic interactions. For these reasons, we have provided as an additional proof of principle of the association of Rab10 to TrkB-positive endosomes, an example of a stationary TrkB-organelle which transiently accumulates Rab10-EGFP for few seconds, just before the onset of processive retrograde transport (Figure 8b). We are actively investigating the mechanism regulating the onset of motion in these organelles, and the molecular events leading to TrkB sorting into retrograde signalling endosomes.

2. Reviewer #1 also indicates that “other than P-CREB immunofluorescence experiments, the authors do not provide data to show the functional effects of Rab10 depletion on BDNF-mediated trophic effects”. While the complexity of retrograde trophic effects of BDNF are well documented [see references 1-3], we have decided to use levels of phosphorylated CREB (pCREB) in the nucleus since activation of this transcription factor constitutes the first node in which the three canonical signalling pathways of TrkB, PI3K-Akt, MAPKs and PLC-γ, converge [see reference 10]. Additionally, the majority of early genes regulated by BDNF depend on the phosphorylation of CREB at serine 133 to increase their expression [see references 19 and 31], making this node a relevant proxy for BDNF signalling and gene response. We have used the signalling readout as a complement to the accumulation of the internalised receptor itself, strengthening the physiological relevance of Rab10-depletion on TrkB transport. The reviewer also makes a very interesting point when focusing on discussion of Rab10 knockdown effect in morphology and survival. They advise that “these data are necessary to strengthen the conclusion that Rab10 is necessary to mediate BDNF-dependent trophic signaling”, adding that we should “assess whether the Rab10 kd or expression of Rab10 DN has general effects on neuronal viability or affect total TrkB levels, and not specifically on retrograde transport of axonally-derived TrkB receptors”. We thank the reviewer for raising these interesting issues. We have now included a new Figure 1, in which we analyse the changes in neuronal density after 24 and 48 hours of Rab10 knockdown. We showed that after 48 hours, there is a significant decrease in survival (Figure 1b), fully justifying our decision of performing this set of experiments between 18 and 24 hours. Importantly, we showed that the levels of endogenous TrkB remain stable even at 48 hours of Rab10 knockdown (Figure 1d), ruling out the depletion of TrkB as a potential confounding factor for the decreased pCREB levels and retrograde TrkB. Please also see point 4 for a discussion about the effects of Rab10 depletion on neuronal morphology. In addition, we followed the suggestion of the reviewer to include “important controls to show the specificity of Rab antibodies, and the knockdown of Rab10 using shRNA. In Figure 1f, the Rab10 levels (assessed by Rab10 immunofluorescence) appear to not be significantly affected by the kd or rescue, which raises questions about the antibody specificity, as well as efficacy of the kd”. In this regard, we have validated the significant decrease of endogenous Rab10 after 24 and 48 hours of treatment with doxycycline (Figure 1c), further optimising our immunofluorescence protocols at the same time.

3. Another interesting point raised by reviewer #1 focused on what happens to the relative concentration of Rab10 and TrkB in the axon. With regard to TrkB, the reviewer points out that: “In Figure 1c (now 2c), the authors show reduced accumulation of axon-derived TrkB receptors in neurons expressing Rab10 shRNA. Images of the axons should be included to assess whether the TrkB receptors accumulate in distal axons, as proposed in the model”. However, no accumulation of TrkB in the periphery is expected upon Rab10 knockdown. Indeed, the population of receptors undergoing retrograde transport is believed to be only a fraction of the total internalised TrkB, with the majority of the receptor engaging in local signalling and recycling back to the plasma membrane [see reference 5]. Moreover, anterograde flux of TrkB in hippocampal neurons has been shown to be regulated by retrograde BDNF signalling [see Zahavi, E. et al. 2021 Dev Cell, DOI: 10.1016/j.devcel.2021.01.010]; therefore, Rab10 knockdown is also expected to decrease the anterograde delivery of TrkB, thus halting any potential distal accumulation of this receptor. We have incorporated these considerations into the model shown in Figure 8a and the Discussion.

Regarding the distribution of Rab10, the reviewer suggests that “it is interesting that BDNF enhances anterograde Rab10 transport based on live imaging. Additional analyses are needed to strengthen the conclusion that Rab10 is enriched in distal axons and to define the mechanisms (phosphorylation?) underlying Rab10 recruitment/retention”. At this point it is important to emphasise that our model does not predict an absolute distal enrichment of Rab10 upon BDNF stimulation either, but a dynamic readjustment on demand of the availability of Rab10 at the site of endocytosis. At any given point, Rab10 carriers are moving anterogradely and retrogradely along the axon. This balance will be transiently shifted towards retrograde or anterograde upon changes in ligand concentration; therefore, the increase in the anterograde bias will proceed only until the demand has been met and it is very unlikely that it results in significant accumulation of Rab10 under physiological conditions. This is precisely why this phenomenon is best appreciated when we perform longitudinal live-cell imaging of axons before and after BDNF stimulation (Figure 5). Regarding the mechanism regulating Rab10 direction bias, we agree with the reviewer about phosphorylation potentially playing a crucial role. We have included this possibility in the discussion, and we have focused our experiments in providing evidence that KIF13B recruitment to Rab10 organelles is increased by BDNF. Following the suggestion of the reviewer that “additional analyses to define mechanisms underlying Rab10 recruitment” and that “the imaging analyses of Kif13B could be complemented with biochemical methods to show BDNF-induced association with Rab10”, we have performed co-immunoprecipitation experiments to show that BDNF significantly increases the interaction between Rab10 and KIF13B (new Figure 6g-h). In line with our interpretation of the imaging analysis in axons, these biochemical data suggest that the effect of BDNF in the recruitment of KIF13B to Rab10 organelles is robust and provides a plausible mechanism for the increase in the anterograde transport of these organelles.

4. Along the same line, both reviewers point out that “the anterograde motor Kif13B is suggested to be involved in BDNF-induced anterograde transport of Rab10 based on confocal microscopy analyses in fixed neurons. However, this aspect is not fully developed. The authors should determine if Kif13B directly interacts with Rab10 and the necessity of Kif13B for anterograde Rab10 transport, retrograde transport of TrkB receptors or BDNF-mediated trophic effects” and “the data with kinesins is not adding much to the understanding of the pathway. No interference with any kinesin is performed”. While the main message of our manuscript is unravelling a new role for axonal Rab10 organelles as dynamic regulators of retrograde TrkB trafficking and signalling, in this amended version of our manuscript, we have presented evidence showing that one specific kinesin, KIF13B, is preferentially recruited to Rab10 organelles upon BDNF stimulation. In stark contrast, the interaction of KIF5B, one of the most abundant axonal kinesins, remains unaffected. We have also responded to the request of confirming the BDNF-mediated interaction between Rab10 and KIF13B with a new set of biochemical experiments (Figure 6g-h).

Although we considered the idea of testing the effects of overexpressing the C-terminus of KIF13B lacking the first 441 residues (KIF13B DN), which includes the motor domain, we reasoned that these experiments may be particularly difficult to adequately control, given that KIF13B has a major role in establishing neuronal polarity [see reference 59]. Our preliminary experiments indeed confirm that the expression of KIF13B DN causes a decrease in morphological complexity that is qualitatively similar to the knockdown of Rab10 (see Author response image 1), further supporting our analysis.

**Author response image 1. sa2fig1:** Comparison of the morphological changes of hippocampal neurons treated with shRNA directed against Rab10 or transfected with a dominant negative version of KIF13B. Neurons in (a) have been transduced with a doxycycline-inducible shRNA system to knock down Rab10 and treated with doxycycline for 48 hours, as indicated in the main Figure 1. In this example, the decrease of Rab10 expression is concomitant with a reduction of dendritic complexity similar to the observed in neurons that have been expressing a motorless mutant of KIF13B (KIF13B DN) for 48 hours (b). Scale bars, 50 µm.

5. Another aspect about which both reviewers had interesting insights and suggestions was the evidence about the role of Rab10 in sorting of TrkB from early endosomes. Reviewer #1 indicates that “the suggestion that Rab10 regulates "sorting" of internalized TrkB receptors is very interesting. The authors show nicely (in Figure 6 [now Figure 7]) that dominant negative Rab10 (rab10DN) does not disrupt TrkB recycling in axons, but increases the co-localization of TrkB with Rab5. The images in Figure 6d [now Figure 7d] also show that Rab5 domains appear to be larger in the Rab10 DN expressing neurons, which is consistent with early endosomes being perturbed. However, additional analyses are needed to define how exactly Rab10 coordinates with Rab5 and Rab7 in "handing off" TrkB from early endosomes to late endosomes in axons” and then adds the following questions: “Which axonal segments were imaged in the analyses in Figure 6? Was this in the distal axons? What happens to TrkB flux in mid-axons and retrograde accumulation in cell bodies in the Rab10 DN or shRNA expressing neurons? Does expression of Rab10 shRNA or Rab10 DN decrease co-localization of retrogradely transported TrkB with Rab7?”. In the same line, reviewer #2 adds that “there is a lot of conjecture about Rab10 vesicles delivering machinery to the distal axon early endosome, but no experiment to address this. Where is TrkB in Rab10 interference conditions? It seems to be stuck with Rab5 and not sorting back to the axonal plasma membrane or into retrograde carriers. Is it accumulating in distal axon tips? It would be good to have live imaging of TrkB in Rab10-DN conditions along axons. The recycling experiments (Figure 6) are very interesting, but it looks like this is happening along the axon shaft. Is this correct?”. We thank to the Reviewers for these comments. Indeed, the confocal images of both experiments shown in Figure 7 show that the expression of dominant-negative Rab10 results in enlarged Rab5-positive early endosomes accumulating internalised TrkB receptor, without affecting its internalisation rate nor its recycling to the plasma membrane. In light of these results, we propose that the increase of TrkB in early endosomes accounts for a specific deficits in its sorting to retrograde signalling endosomes. These experiments were done in mass cultures and mid-axons were imaged, since they represented the more isolated segments allowing analysis at the level of individual axons, something that is rarely possible in microfluidic chambers, where axons are bundled together within microgrooves. BDNF was added to the media in these cultures, therefore, both internalisation and recycling are observed along the entire axon and not uniquely at axon terminals. Interestingly, it is under these same exact conditions that the live-cell imaging of Rab10EGFP (Figure 5) has been done, showing that the BDNF-induced increase in net anterograde transport is not a consequence of a gradient of ligand, but an intrinsic feature of the system which we propose is the result from an increased interaction of Rab10 organelles with the anterograde motor KIF13B. In our model (Figure 8a), we have integrated the fact that the main source of BDNF in the nervous system are synaptic targets; therefore, this mechanism would operate mobilising Rab10 organelles towards nerve terminals or retaining them at TrkB internalisation hotspots until the system reaches equilibrium.

Regarding the hypothesis that Rab10 organelles deliver molecules essential for sorting to the distal axon, it is important to notice that Rab10 has been shown to recruit TBC-2 to the membrane of early endosomes, thus regulating the sorting of cargoes towards basolateral recycling [see reference 57]. Another additional example is the cooperation between Rab10 and Rab5 for the sorting of dense core vesicles in the Golgi apparatus [see Sasidharan, N. et al. 2012, PNAS, DOI: 10.1073/pnas.1203306109]. Interestingly, one of the best characterised roles of KIF13B is the anterograde transport of phosphoinositide-3,4,5-triphosphate (PIP_3_), which is crucial for the regulation of Akt signalling and endosomal maturation [see reference 58]. In *Drosophila*, the orthologue of KIF13B, Khc-73, has been also involved as a regulator of BMP retrograde signalling carriers [see reference 61]. As reviewer #2 pointed out (“Are the anterograde Rab10 carriers endosomally derived? It would be important to characterize the cargos of these carriers”), a more detailed and unbiased analysis of the composition of the pool of anterograde Rab10-positive organelles is warranted to identify specific components of the sorting machinery.

Although we agree with the Reviewers when predicting of a decreased interaction between TrkB and Rab7 in the absence of functional Rab10, we believe that this experiment would not add to the model, considering that we have already shown an increased accumulation of TrkB in Rab5 domains in the absence of functional Rab10 (Figure 7), and a significant decrease in retrograde TrkB delivered to the soma (Figure 2). The transition between a Rab5-Rab10 hybrid domain to Rab7-positive membranes (or *“handing off”*) fits with the almost complete exclusion between Rab10 and Rab7 that we show in Figure 3a-b. However, it is important to note that not all retrograde signalling carriers are Rab7-positive, and signalling endosomes have been shown to be heterogenous in nature, including Rab5-positive compartments moving in a less processive manner in the retrograde direction and carrying signalling molecules for long distances [see references 12 and 21]. Indeed, neurons expressing dominant-negative Rab10 display less processive retrograde transport of internalised TrkB compared to neurons transduced with wildtype Rab10 (see Author response image 2). Additionally, in Figure 8b, we have provided an interesting example of transient increase of Rab10 in a TrkB-containing organelle right after its transition between stationary and retrograde, which points in the same direction. These data and references have been included in the discussion of the revised manuscript.

**Author response image 2. sa2fig2:** Retrograde transport of TrkB in neurons expressing Rab10 DN. Using the same methods illustrated in Figure 4ad, TrkB-Flag was internalised and axons of neurons expressing Rab10 DN were imaged. (a) Example of a retrograde TrkB carrier. (b) Kymograph showing several cargoes with different degrees of processivity and pausing frequency.

6. Reviewer #2 suggested “to use super-resolution imaging to define the morphology of TrkB/Rab10-positive endosomes in distal axons”, considering that “Rab10 has been proposed to play a role in generating tubular endosomes”. We thank to the reviewer for this suggestion. We have included in the revised manuscript Supplementary Video 3, showing examples of tubular endosomes in the cell body of a living neuron. This video, which was recorded by using high resolution Airyscan confocal microscopy, clearly shows that the morphology of the Rab10 organelles is indeed very diverse

7. Reviewer #2 asked us to briefly explain the relationship between what we have found in this work and the mass spec analysis from signalling endosomes in Debaisieux et al. 2016 [see reference 18]: “Rab10 was enriched in SEs by SILAC/mass spec (from a previous paper by this group) but now is not in the retrograde carriers. Can you discuss?”. In this work, we have used the binding fragment of tetanus toxin (H_c_T) to isolate organelles containing this cargo from mouse stem cells-derived motor neurons after 10, 30 and 60 minutes of endocytosis. Whilst organelles isolated at early time points are enriched in endocytic and early endosomal markers, fractions isolated at later time points are positive for late endosomal markers. Hence, the abundance of Rab7 increase steadily at 30 and 60 min as shown in the figure below (see Author response image 3). In contrast, Rab10 first decreases and then remain constant at 30 to 60 mins, suggesting that it is not accumulated on these organelles during signalling endosome maturation. These data fit our confocal microscopy analysis using in hippocampal neurons shown in Figure 3e-f.

**Author response image 3. sa2fig3:** Rab10 and Rab7 show different recruitment behaviour in signalling endosomes. Cross correlation of the enrichment of >2,000 proteins detected from immunoisolated HcT-containing signalling endosomes purified from mouse stem cell-derived motor neurons. Rab7 has been highlighted as example of a protein that is enriched at later time points, whereas Rab10 is preferentially associated to early compartments. Modified from Debaisieux et al. 2016 [ref. 18].

8. Reviewer #2 noted that “It is not always clear what experiments are done in microfluidic chambers. Please specify in each figure”. We apologise for the confusion. We have now indicated in the revised manuscript when experiments have been performed in microfluidic chambers. When not indicated, they have been done in mass culture. The reviewer adds: “I think the observation that BDNF affects the directionality of the Rab10 carriers is very intriguing. Were these in microfluidic chambers? How is BDNF signaling conveyed to Rab10 carriers?”. As previously explained (see point 5), Rab10-EGFP live-cell imaging have been done in mass culture, so BDNF signalling can be acting along the entire axon. However, preliminary experiments in microfluidic chambers have shown that the same change in the direction bias occur when BDNF is added only to the terminals, suggesting that retrograde propagation of BDNF signalling is sufficient to inform Rab10-positive organelles in transit. Whilst the exact mechanism is not yet clear, coupling or merge of Rab10 organelles may facilitate the anterograde mobilisation of this compartment, as exemplified by the new Supplementary video 2, where a small stationary Rab10-positive organelle merge with a larger mobile compartment and the resulting Rab10positive combined compartment continues its anterograde motion.

9. Reviewer #2 also asked the following questions: “Is p75 not in a complex with TrkB after BDNF binding? What is the interpretation/implication of p75 being in different carriers in the axon? I find this a very interesting observation, but the relevance is not explained or further explored”. We thank the reviewer for highlighting this interesting aspect of the previous Figure 4. Whereas it is not entirely surprising that TrkB and p75 can be found in different organelles given their differential mechanisms of internalisation and post-endocytic dynamics; the presence of p75 alone in anterograde Rab10 carriers results intriguing. Bruce Carter’s group has previously shown that p75 is mobilised to the plasma membrane following TrkA activation with NGF in sympathetic neurons, which results in stronger TrkA signalling capabilities [see reference 55]. They show that this mobilisation of p75 depends on the activation of Arf6. Interestingly, in *C. elegans* it has been shown that Rab10 and Arf6 reside in the same subset of organelles. Whether concentration of p75 receptor in different domains of the axon is regulated by Rab10 organelles is an interesting possibility that we will be excited to explore in the future.

10. Reviewer 2# has pointed out that “all the discussion of LRRK2 and phosphorylation is besides the point since the authors do not test involvement of this regulatory mechanism”. We agree with the reviewer, and therefore, the discussion of this aspects has been summarised and focused on Rab10-phosphorylarion as a potential regulatory mechanism for the recruitment of motors and other effectors.

[Editors’ note: what follows is the authors’ response to the second round of review.]

We thank you and the reviewers for their positive comments and very insightful suggestions during the second round of revisions of our manuscript titled “Rab10 regulates the sorting of internalised TrkB for retrograde axonal transport”. We were delighted to learn that Reviewer #1 concerns were fully satisfied, and very grateful for the advice from Reviewers #2 and #3, who not only recognised the quality and interest of our work, but also helped us to further distil the key messages of our manuscript. Their comments prompted us to add extra controls and attempt new experiments, which made our work more robust and interesting for the readership of *eLife*.

Three main questions persisted from Reviewers #2 and #3; they are all addressed in this resubmission. The first concern was about the nature of the Rab10 axonal compartments, and whether they correspond to previously described recycling endosomes regulated by other Rab GTPases; the other two questions both focussed on the specificity of KIF13B as the kinesin mobilising anterograde axonal Rab10. In this revised version, we present novel evidence addressing these matters and other minor points raised by the Reviewers. At the same time, we were pleased to see that none of the questions concerned the key findings of our work: a novel mechanism by which BDNF regulates Rab10 anterograde delivery, and Rab10 mediates sorting of internalised TrkB to the retrograde axonal transport route.

Specifically,

1. Reviewers #2 and #3 agreed when saying that “the fact that there is little colocalization between Rab10 and TrkB needs further characterization”. Additionally, Reviewer #2 suggested an alternative interpretation of our data suggesting a model prescinding of a transient association of TrkB and Rab10 in axons. In figure 3, we show both by quantitative confocal microscopy (figure 3e) and by using super-resolution radial fluctuations (figure 3f) that after 30, 60 and 90 minutes upon BDNF treatment, endogenous retrograde TrkB and Rab10 partially co-localise, and this colocalisation does not increase with time. We also showed that this association can be stabilised if both proteins are overexpressed, so it can be also observed as events of retrograde co-transport (figure 4a-d). Moreover, in the last figure we documented an example where, even under overexpression conditions, the association between a TrkB-containing signalling endosome and Rab10 was still transient and linked to the processivity of the organelle (figure 8b). It is also noteworthy that Rab10 knock-down caused a significant decrease of retrograde transport of endogenous TrkB (figure 2d-e), further stressing the functional relevance of this transient association. Promted by the reviewers asking for additional proofs of this association, we attempted an unbiased proteomic approach based on the immunoisolation of Rab10-positive membranes from primary cortical neurons by using antibody-conjugated magnetic beads followed by mass spectrometry. Using this approach, we confirmed that TrkB is present in Rab10containing membranes. We are currently optimising the immunoisolation approach, and the full results will be made available in a follow-up study. Thus, we have used four different experimental setups to confirm that TrkB transiently associates with Rab10-containing organelles.

2. Along the same lines, Reviewers #2 and #3 pointed that the low level of co-localisation “is not consistent with "a new Rab10 organelle" that mediates TrkB sorting and retrograde transport”. We fully agree with this conclusion and we find very important to clarify that what we have called “Rab10 organelles” can be more precisely described as membrane domains enriched in Rab10, which eventually interact with the endo-lysosomal system. The use of a more precise description of the compartment is particularly important in light of the diverse distribution of Rab10 in multiple organelles and membrane subdomains that has been observed in different cell types [Chua, C. E. L., and Tang, B. L. (2018). Rab 10-a traffic controller in multiple cellular pathways and locations. *J Cell Physiol, 233*, 6483-94. https://doi.org/10.1002/jcp.26503].

We, therefore, have changed all the references to “Rab10 organelles” into “Rab10 compartments”, which better reflects the potentially heterogeneous nature of the Rab10-containing membranes in the axon. Regarding this aspect, Reviewers also pointed to Rab10 being related to Rab11 and recycling, so they suggest to “test other recycling endosome Rabs, such as Rab11, Rab14, Rab10 to determine if these are really novel organelles or simply Rab11-recycling endosomes that also contain Rab10”. Although Rab10 and Rab11 do not belong to structurally related groups of Rabs, they have been shown to converge in a sub-class of recycling endosomes [Gupta, K., Mukherjee, S., Sen, S., and Sonawane, M. (2022). Coordinated activities of Myosin Vb isoforms and mTOR signaling regulate epithelial cell morphology during development. *Development, 149*, https://doi.org/10.1242/dev.199363; Homma, Y., and Fukuda, M. (2016). Rabin8 regulates neurite outgrowth in both GEF activity-dependent and independent manners. *Mol Biol Cell, 27*, 2107-18. https://doi.org/10.1091/mbc.E16-02-0091]. However, there are several reasons why it is unlikely that Rab10-positive compartments we observe moving in the axon are Rab11-containing recycling endosomes. First, Rab11 organelles have been shown to be infrequent in the axon of hippocampal neurons, while we and others observe a large number of Rab10-positive axonal puncta. Second, blocking Rab10 function by overexpressing a dominant negative mutant (T23N) did not alter TrkB recycling in the axon. However, we fully agree with the reviewers regarding the need of a better characterisation of the Rab GTPases that are present in the Rab10-positive axonal membranes. Our unbiased proteomic characterisation of immunoisolated Rab10 compartments revealed the presence of more than 30 different Rabs, including those associated with the endolysosomal system (Rab4, Rab5, Rab7, Rab11). Importantly, because our starting material for immunoisolation was not pure axonal membranes, we decided to assess the co-localisation of endogenous Rab10 and other relevant Rabs by immunofluorescence. Confocal microscopy analyses confirmed a significant increase of Rab10 on Rab5 domains upon stimulation with BDNF, which has been added to the revised figure 3 (panel c).

To address the question of Reviewer #2, we have also analysed the co-localisation between Rab10 and the main Rabs associated with recycling endosomes, Rab4 and Rab11. These experiments were performed by using the same approaches previously described in the manuscript. Axon segments from three independent experiments were analysed. The data suggest that around 25% of Rab10-positive compartments also contained these slow recycling endosome markers. As shown on the left, the amount of Rab10 co-localising with Rab4 and Rab11 exhibited no changes upon BDNF treatment. However, the amount of Rab11 in Rab10-positive membranes increased after 30 minutes of BDNF treatment, suggesting that the distribution of the minute amount of Rab11 present in the axon is regulated by neurotrophic signalling and eventually converges into the same pool of Rab10-positive organelles. This is consistent with a scenario in which, at steady state, Rab10 is found in diverse axonal compartments, and upon treatment with BDNF, it relocalises to early endosomes positive for Rab5 and other endosomal Rabs.

3. Reviewer #2 raised questions about the reasons for studying KIF5B and KIF13B. Together with Reviewer #3, they suggested to analyse the effects of kinesin down-regulation on the transport of TrkB. In this regard, it is important to mention that we have not aimed to study direct effects of these or other kinesins on the axonal transport of TrkB, but to propose a mechanism for the BDNF-dependent regulation of Rab10 direction bias, the dynamics of which is, in turn, crucial to fine-tune TrkB sorting from early endosomes. However, it is important to notice that these kinesins have many other roles and, therefore, the prediction is that knocking them down will cause disruption of axonal dynamics of multiple cargoes. For example, anterograde transport of TrkB has been shown to be regulated by kinesin 1 [Huang, S. H., Duan, S., Sun, T., Wang, J., Zhao, L., Geng, Z., Yan, J., Sun, H. J., & Chen, Z. Y. (2011). JIP3 mediates TrkB axonal anterograde transport and enhances BDNF signaling by directly bridging TrkB with kinesin-1. *J Neurosci*, 31, 10602-14. https://doi.org/10.1523/JNEUROSCI.0436-11.2011; Sun, T., Li, Y., Li, T., Ma, H., Guo, Y., Jiang, X., Hou, M., Huang, S., & Chen, Z. (2017). JIP1 and JIP3 cooperate to mediate TrkB anterograde axonal transport by activating kinesin-1. *Cell Mol Life Sci*, 74, 4027-44. https://doi.org/10.1007/s00018-017-2568-z; Zahavi, E. E., Hummel, J. J. A., Han, Y., Bar, C., Stucchi, R., Altelaar, M., & Hoogenraad, C. C. (2021). Combined kinesin-1 and kinesin-3 activity drives axonal trafficking of TrkB receptors in Rab6 carriers. *Dev Cell*, 56, 494-508 e497. https://doi.org/10.1016/j.devcel.2021.01.010]; hence, the effects of KIF5B knock down will be necessarily complex and not only affect Rab10 trafficking. Similarly, KIF13B is crucial for establishing and maintaining neuronal polarity; therefore, knocking it down would also disrupt axon integrity [Nakata, T., & Hirokawa, N. (2007). Neuronal polarity and the kinesin superfamily proteins. *Sci STKE*, 2007, pe6. https://doi.org/10.1126/stke.3722007pe6]. Previously, we presented the results of a pilot experiment, in which we attempted to use a KIF13B mutant lacking the motor domain, which yielded a significant decrease in neurite complexity, emphasising the need to develop new tools to further explore this point. Instead, we attempted this experiment for a different reason: we wondered which interactors of Rab10 mediate its differential direction bias in response to BDNF. Since their interaction is regulated by phosphorylation, we initially focussed on the adaptors JIP3 and JIP4; however, we found little co-localisation between JIP3, JIP4 and Rab10, as documented in the supplementary material. We then looked at KIF13B because it was shown to preferentially interact with GTP-bound Rab10, and it is highly expressed in the axon of hippocampal neurons [Yang, R., Bostick, Z., Garbouchian, A., Luisi, J., Banker, G., & Bentley, M. (2019). A novel strategy to visualize vesicle-bound kinesins reveals the diversity of kinesin-mediated transport. *Traffic*, 20, 851-66. https://doi.org/10.1111/tra.12692]. KIF5B was chosen as control, because it is also robustly expressed in the axon of hippocampal neurons but lacks a binding site for Rab10. It is relevant to mention that, in contrast to Reviewer #2’s views, stating that “the change in Kif13B and Kif5B intensity in response to BDNF is very moderate at best”, we don’t focus on the change of intensity of the KIFs in the axon, but on their recruitment to Rab10positive domains. While KIF13B increased in Rab10 compartments compared to the rest of the cytoplasm, KIF5B displayed the opposite behaviour (figure 6b-e). As now explained in the Discussion, it is possible that this is not the only determinant of Rab10 direction bias, but the contribution of KIF13B would be enough to explain the BDNF-dependent changes of direction, providing a lead for further studies on the detailed mechanism controlling Rab10 dynamics.

4. Related to the previous point, Reviewers #2 and #3 also expressed concerns about the immunoprecipitation experiment shown in figure 6g-h. They mentioned that “it appears that there is more HA-Rab10 precipitated in BDNF-treated samples, thus, it is unclear whether slight increase in GFP-Kif13B co-precipitation is due to that rather than increased interaction with Rab10”. Upon quantification, we are pleased to report that there is a two-fold increase in Kif13B (p = 0.0024). The reviewer is absolutely right at noticing that a little more HA-Rab10 immunoprecipitated in the BDNF-treated sample in figure 6g; however, figure 6h displays the ratio between GFP-KIF13B and HA-Rab10 signals, which normalises data for variations in the efficiency of the immunoprecipitation. To better clarify this point, we have now included the quantification of the non-normalised GFP-KIF13B (showing a two-fold increase with a p value = 0.0067), and the quantification of the HA-Rab10 showing no differences (p value = 0.9240).

The reviewer also questions the presence of a slight band of GFP-KIF13B in the sample that does not expressed HA-Rab10. Indeed, a small amount of the GFP fusion protein is nonspecifically bound to the magnetic beads, which is a common occurrence in this type of experiments, yet the low intensity of this band does not affect the overall interpretation of our results.

5. The aim of the immunoprecipitation experiments was to show (i) that KIF13B and Rab10 can interact as part of a molecular complex, and (ii) that the interaction was positively regulated by BDNF. Both aspects were indisputably proven by the results shown in figure 6g-h, complementing the spatial correlation reported in figure 6a-c. However, the Reviewer #2 asked whether we could “blot for endogenous Kif13B co-precipitation with HA-Rab10”.

We did several pulldown experiments from Neuro 2A cells transfected with HA-Rab10. Although we were able to detect endogenous KIF13B in the input, we failed to reveal KIF13B in the immunoprecipitate (see Author response image 4). A possible explanation for this negative result might derive from a different localisation and/or expression of KIF13B in Neuro 2A cells compare to primary neurons. Using a lentiviral inducible system to efficiently express mCherry-Rab10 in hippocampal neurons, we used magnetic beads to immunoprecipitate mCherry-Rab10, but we again failed to observe endogenous KIF13B associated to the beads under our experimental conditions. Whilst disappointing, this result is hardly surprising, since our total lysates are not enriched in axonal membranes, and we did not deploy any effective mean to stabilise the transient interaction between GTP-bound Rab10 and KIF13B. We believe that determining the content of Rab10-positive membrane compartments by using immunoisolation and proteomics will provide a complete and unbiased characterisation of this Rab10 compartment; however, we hope that the Reviewers agreed with us that this aim goes beyond the focus of this manuscript.

**Author response image 4. sa2fig4:** 

6. Reviewer #2 also raised the potential confounding effects of the Rab10 DN, which they suggested to mitigate by repeating all recycling experiments in neurons treated with shRNA for Rab10 (“all recycling experiments need to be done in Rab10-KD cells”). Far from showing confounding effects, our results revealed no effect of Rab10 DN in recycling of TrkB in axons. However, expression of Rab10 DN effectively induced an increase on TrkB/Rab5 colocalisation in axons, which is consistent with the reduction on retrograde accumulation observed in Rab10-KD neurons. On the same issue, Reviewer #2 argues that “it is very puzzling that trapping TrkB in early endosomes did not affect recycling. Most plasma membrane receptors recycle by sequential transport from early endosomes to recycling endosomes. Consequently, I would expect that trapping TrkB in early endosomes would decrease its recycling” We apologise for the lack of clarity explaining our model. The key assumption in this point of the reviewer is that TrkB receptors located in the early endosome are homogeneously distributed and equally available to be destined either to local recycling or retrograde trafficking. However, the sorting endosome is rich in membrane subdomains and this heterogeneity has been hypothesised to be crucial for cargo selection and trafficking control [Redpath, G.M.I, Betzler, V.M, Rosatti, P. and Rossy, J., (2020). Membrane Heterogeneity Controls Cellular Endocyctic Trafficking. Front. *CellDev. Biol.*, 8, 757. https://doi.org/10.3389/fcell.2020.00757]. If retrograde and recycling pools of TrkB were differentially regulated in early endosomes, it is expected that blocking the sorting to retrograde transport would increase colocalisation of TrkB and Rab5, with no increase on recycling. Several lines of evidence indicate the potential of Rab10 and its effectors to define membrane sub-domains important for cargo selection and regulation of membrane curvature, budding and formation of tubules. We have included and discussed this literature in the manuscript, as well as rephrased any ambiguity on the interpretation of these experiment.

7. Finally, Reviewer #1 also provided very useful comments about the nomenclature of Rab10 constructs, a couple of missing resources in the table and asked for clarification about a reference. We have amended the manuscript accordingly.